# The Role of *luxS* in *Campylobacter jejuni* Beyond Intercellular Signaling

Dina Ramić,[a] Blaž Jug,[a] Katarina Šimunović,[a,b] Magda Tušek Žnidarič,[c] Urban Kunej,[d] Nataša Toplak,[e] Minka Kovač,[e] Marjorie Fournier,[f] Polona Jamnik,[a] Sonja Smole Možina,[a] Anja Klančnik[a]

[a]Department of Food Science and Technology, Biotechnical Faculty, University of Ljubljana, Ljubljana, Slovenia
[b]Department of Microbiology, Biotechnical Faculty, University of Ljubljana, Ljubljana, Slovenia
[c]Department of Biotechnology and System Biology, National institute of Biology, Ljubljana, Slovenia
[d]Department of Agronomy, Biotechnical Faculty, University of Ljubljana, Ljubljana, Slovenia
[e]Omega, d.o.o., Ljubljana, Slovenia
[f]Sir William Dunn School of Pathology, University of Oxford, Oxford, United Kingdom

**ABSTRACT** The full role of the *luxS* gene in the biological processes, such as essential amino acid synthesis, nitrogen and pyruvate metabolism, and flagellar assembly, of *Campylobacter jejuni* has not been clearly described to date. Therefore, in this study, we used a comprehensive approach at the cellular and molecular levels, including transcriptomics and proteomics, to investigate the key role of the *luxS* gene and compared *C. jejuni* 11168Δ*luxS* (*luxS* mutant) and *C. jejuni* NCTC 11168 (wild type) strains. Transcriptomic analysis of the *luxS* mutant grown under optimal conditions revealed upregulation of *luxS* mutant metabolic pathways when normalized to wild type, including oxidative phosphorylation, carbon metabolism, citrate cycle, biosynthesis of secondary metabolites, and biosynthesis of various essential amino acids. Interestingly, induction of these metabolic pathways was also confirmed by proteomic analysis, indicating their important role in energy production and the growth of *C. jejuni*. In addition, genes important for the stress response of *C. jejuni*, including nutrient starvation and oxidative stress, were upregulated. This was also evident in the better survival of the *luxS* mutant under starvation conditions than the wild type. At the molecular level, we confirmed that metabolic pathways were upregulated under optimal conditions in the *luxS* mutant, including those important for the biosynthesis of several essential amino acids. This also modulated the utilization of various carbon and nitrogen sources, as determined by Biolog phenotype microarray analysis. In summary, transcriptomic and proteomic analysis revealed key biological differences in tricarboxylic acid (TCA) cycle, pyruvate, nitrogen, and thiamine metabolism as well as lipopolysaccharide biosynthesis in the *luxS* mutant.

**IMPORTANCE** *Campylobacter jejuni* is the world's leading foodborne bacterial pathogen of gastrointestinal disease in humans. *C. jejuni* is a fastidious but widespread organism and the most frequently reported zoonotic pathogen in the European Union since 2005. This led us to believe that *C. jejuni*, which is highly sensitive to stress factors (starvation and oxygen concentration) and has a low growth rate, benefits significantly from the *luxS* gene. The role of this gene in the life cycle of *C. jejuni* is well known, and the expression of *luxS* regulates many phenotypes, including motility, biofilm formation, host colonization, virulence, autoagglutination, cellular adherence and invasion, oxidative stress, and chemotaxis. Surprisingly, this study confirmed for the first time that the deletion of the *luxS* gene strongly affects the central metabolic pathway of *C. jejuni*, which improves its survival, showing its role beyond the intercellular signaling system.

**KEYWORDS** *Campylobacter jejuni*, *luxS*, intercellular signaling, transcriptome, proteome, metabolism, stress response

Address correspondence to Anja Klančnik, anja.klancnik@bf.uni-lj.si.

The authors declare no conflict of interest.

*C*ampylobacter spp. are the leading bacterial cause of human gastroenteritis worldwide, and they have been the most frequently reported foodborne pathogen in the European Union since 2005, causing health and economic damages with associated costs of €2.4 billion per year (1, 2). *Campylobacter jejuni* is a Gram-negative, microaerophilic commensal of many wild and food-associated animals that are destined for human consumption (3, 4). During processing, *C. jejuni* is released from the guts of the animal hosts, contaminating meat products and surfaces. The consumption of undercooked contaminated meat can cause campylobacteriosis, with symptoms ranging from watery to bloody diarrhea (1, 5). *C. jejuni* is an important bacterial species due to its high incidence in the food industry and public health system, the duration of campylobacteriosis, possible complications, such as Guillain-Barré syndrome, and the wide distribution of multidrug-resistant *C. jejuni* strains. These issues have an impact not only on food safety and health care but also on socioeconomics (6, 7). *Campylobacter* spp. are responsible for over 166 billion gastrointestinal diseases and approximately 38,000 deaths per year worldwide (2). Moreover, although *C. jejuni* is difficult to cultivate in the laboratory, it is a widespread and persistent bacteria in the food industry; this is known as the "*Campylobacter* paradox" (8). In addition, temperature fluctuations, oxidative stress, starvation, and other stress factors present in the food-processing environment trigger the transition to a viable but nonculturable (VBNC) state, which correlates with prolonged persistence of *Campylobacter* in the food chain. In this state, *C. jejuni* can still infect humans (9, 10). Understanding metabolic changes is part of the basic knowledge about the life cycle of *C. jejuni* and will help in the fight against this leading foodborne pathogen, ensuring better safety of food products. Because the gene *luxS* is reported to be upregulated in surviving cells at low temperatures in chicken juice, a common food model for the natural habitat of *C. jejuni*, and its absence leads to reduced colonization and biofilm formation (11, 12), in this study, we focused on evaluating its involvement in the *C. jejuni* life cycle.

Elvers and Park (13) were the first to describe the presence of *luxS* S-ribosyl homocysteinase, encoded by the *luxS* gene, in *C. jejuni* NCTC 11168. Its primary function in bacterial cells is in the *S*-adenosyl-homocysteine recycling pathway, where it is responsible for the hydrolysis of *S*-adenosylhomocysteine to homocysteine, which is then further metabolized to *S*-adenosylmethionine (SAM) (14). The SAM pathway is included in methyl recycling in bacteria, which is associated with bacterial DNA methylation, chemotaxis, motility, and different metabolic and biosynthetic reactions. The SAM pathway is also crucial for bacterial polyamine formation and vitamin synthesis (15). Alteration of the SAM pathway associated with *luxS* mutagenesis can have significant impacts on bacterial metabolism. One of the products of the SAM pathway is 4,5-dihidroxy-2,3-pentandione (DPD), which is spontaneously cyclized to form AI-2, an interspecies signaling molecule (16). We have previously shown that the concentration of AI-2 in *C. jejuni* increases linearly with the increasing of cell concentration, which suggests that this signaling molecule is only a byproduct of the methyl cycle (17). This has also focused our current work to determine the role of *luxS* using molecular methods.

In *C. jejuni*, the transcription of the *luxS* gene has been associated with several phenotypes, including intercellular signaling, motility, biofilm formation, host colonization, virulence, autoagglutination, cellular adherence and invasion, oxidative stress, and chemotaxis (Fig. 1) (13, 16, 18–23). It has been shown that the intercellular signaling is completely absent in the *luxS* mutant, while motility, biofilm formation, autoagglutination, host colonization, and invasion of the *luxS* mutant are decreased in comparison to the wild type (13, 16, 20–26).

Despite it being clear that the *luxS* gene is important for many phenotypes of *C. jejuni*, the role of *luxS* in other biological processes, such as synthesis of different essential amino acids and nitrogen and pyruvate metabolism, has not been clearly described to date. As such, we used a comprehensive approach at different levels, including physiology, transcriptomics, and proteomics, to investigate the key role of the *luxS* gene, comparing *C. jejuni* 11168Δ*luxS* (*luxS* mutant) to *C. jejuni* NCTC 11168 (wild type). First, the *luxS*-mutant

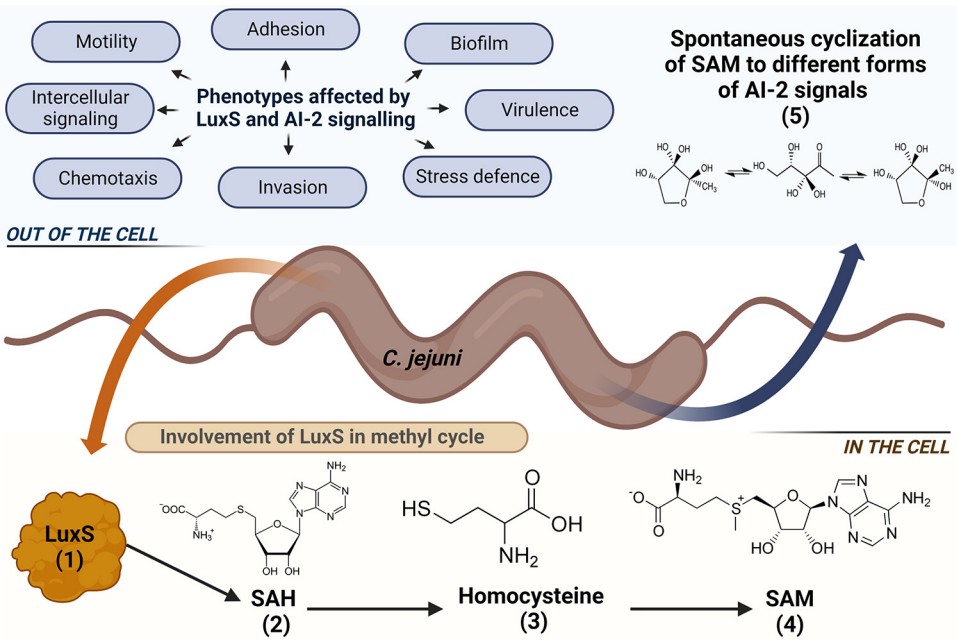

**FIG 1** Schematic representation of the role of LuxS protein in *C. jejuni*. The main function of LuxS in bacterial cells is the *S*-adenosyl-homocysteine recycling pathway (1), which is part of the methyl cycle, where it is responsible for the hydrolysis of *S*-adenosylhomocysteine (SAH; 2) to homocysteine (3), which is then further metabolized to SAM (4). One of the products of the SAM pathway is also 4,5-dihidroxy-2,3-pentanedione (DPD), which is spontaneously cyclized to AI-2, an interspecies signaling molecule (5). Phenotypes affected by *luxS* and AI-2 signaling in *C. jejuni* include intercellular signaling, motility, adhesion, biofilm formation, chemotaxis, invasion, virulence, and stress defense.

and wild-type strains were used to assess survival and morphological differences under stress conditions (starvation and oxidative stress). Their utilization of different carbon and nitrogen sources was also investigated. Further, we performed next-generation sequencing to gain insight into transcriptomic differences between the *luxS*-mutant and wild-type strains. In addition, label-free relative quantitative mass spectrometry was used for proteomic analysis of the selected strains. This article represents the first attempt to investigate the involvement of the *luxS* gene in *C. jejuni* processes at the molecular level.

## RESULTS

**Survival of the *luxS*-mutant and wild-type strains under stress conditions.** *luxS*-mutant and wild-type *C. jejuni* were cultivated under optimal conditions (optimal atmosphere, medium, and temperature) to determine basic differences that might arise during their growth (27). Further, survival of the *luxS*-mutant and wild-type strains was studied also under starvation (low-nutrient) and oxidative stress (aerobic atmosphere) conditions. There were no significant differences in the survival of either strain under optimal conditions during the whole period of cultivation ($P \geq 0.05$) (Fig. 2A; Fig. S1 in the supplemental material). The wild type was more sensitive to starvation ($P \leq 0.05$) than the *luxS* mutant at each time point (Fig. 2B). These differences were more than 3 $\log_{10}$ CFU/mL. Oxidative stress had no significant effect on the survival of either strain, except after 120 h, where the survival of the wild type was 1 $\log_{10}$ CFU/mL higher than the *luxS* mutant ($P \leq 0.05$) (Fig. 2C).

Negative-staining preparation of biological samples for transmission electron microscopy (TEM) provides information on bacterial shape and size, and we followed the effects of starvation and the presence of oxygen on morphology of the *C. jejuni* wild-type and *luxS*-mutant strains. Under normal growing conditions, the *C. jejuni* wild type and *luxS* mutant displayed a mixture of two bacterial morphotypes: helical (Fig. 3A) and a rod to curvature shape (Fig. 3B) with two flagella, one on each side. Although the helical shape is described as the normal shape of *C. jejuni*, their morphology can be transformed into a rod shape. Both shapes were present in both strains, but in the *luxS* mutant, the rod shape prevailed, and the cells were

**FIG 2** (A to C) Growth rates of *C. jejuni* NCTC 11168 and *C. jejuni* 11168Δ*luxS* (*luxS* mutant) during cultivation under optimal (A), starvation (B), and oxidative (C) conditions at 42°C during 120 h of incubation. Data are average $\log_{10}$ CFU values and are presented as means ± standard deviation from three replicates; *, $P \leq 0.05$ versus relevant control (one-way analysis of variance [ANOVA] with Tukey's *post hoc* tests).

smaller (1.96 $\mu$m × 0.67 $\mu$m) compared with the wild type (2.77 $\mu$m × 0.61 $\mu$m). Starvation and oxidative stress triggered the same response in both strains. The helical and curvature shapes were transformed into a coccoid form (Fig. 3C) under both stress conditions, and after 5 days of exposure, we could only find the coccoid form. Immediately (4 h) after exposure to oxygen stress, we could not find any preserved cells in the sample of the *luxS* mutant; only the coccoid morphology was present.

**Utilization of different carbon and nitrogen sources by the *luxS* mutant and wild type.** Results obtained with the Biolog system showed that the *luxS* mutant

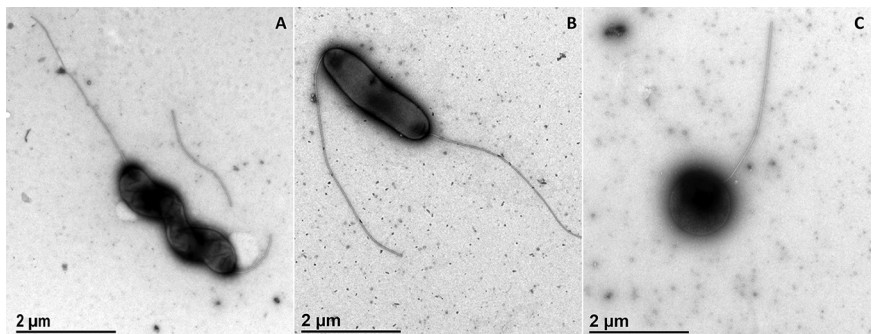

**FIG 3** Representative micrographs of *C. jejuni* morphology. (A) Spiral shape, *C. jejuni* NCTC 11168. (B) Rod shape, *C. jejuni* 11168Δ*luxS* (*luxS* mutant). (C) Coccoid shape, *C. jejuni* 11168Δ*luxS* (*luxS* mutant); scale bars, 1 $\mu$m.

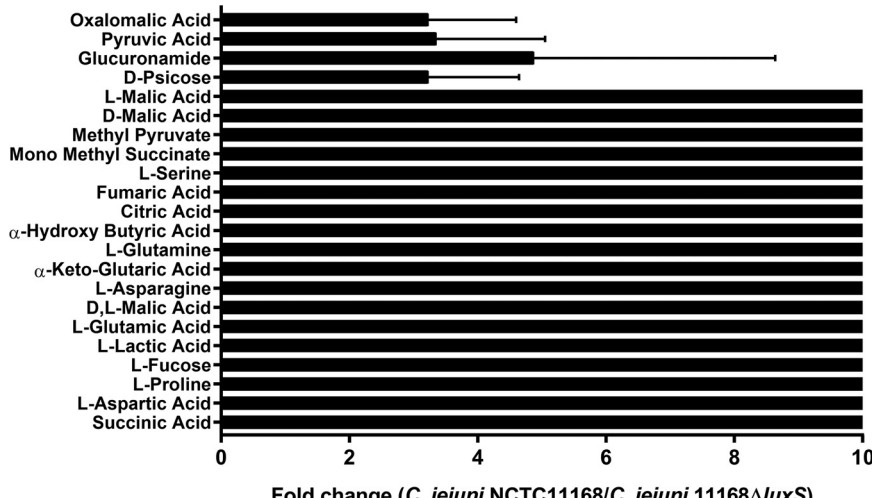

**FIG 4** Differences between *C. jejuni* NCTC 11168 and *C. jejuni* 11168Δ*luxS* (*luxS* mutant) utilization of different carbon and nitrogen sources measured with the Biolog system. Data are presented as fold change of signal gathered from wild type compared to the *luxS* mutant.

poorly metabolized only 22 of 288 different carbon and nitrogen sources compared to the wild type ($P \leq 0.05$) (Fig. 4). The signal collected from the wild type was 10 times higher than that from the *luxS* mutant when exposed to L-malic acid, L-proline, L-serine, fumaric acid, citric acid, L-glutamine, L-asparagine, L-fucose, and others, implying that the metabolism of the *luxS* mutant was limited after exposure to these nutrients ($P \leq 0.05$). Furthermore, this means that the growth rate and survival ability of the *luxS* mutant on these sources are lower than those of the wild type, which grew better than the *luxS* mutant on these 22 sources.

**Transcriptomic profile of the *luxS* mutant and wild type.** Comparing the transcriptomes of wild-type and *luxS*-mutant *C. jejuni* can improve our understanding of the role of the *luxS* gene in the *C. jejuni* life cycle. Strains were cultivated until the middle of the exponential phase, during which numerous changes in gene expression are expected.

Detailed analysis showed that 765 genes were differentially expressed in the *luxS* mutant when normalized to the wild type (false-discovery rate [FDR], $P \leq 0.05$) after 16 h of incubation. Among these genes, 354 were upregulated, and 402 were downregulated (Fig. 5; Table S1). However, the list (Table S1) of these transcripts was selected because they significantly differed between the *luxS* mutant and wild type (log$_2$ fold change [FC] of $\geq 1$, $P \leq 0.05$). Some of these genes are listed below because they can explain our observations at the physiological level, such as growth under starvation-like conditions. Among the upregulated genes, genes important for *C. jejuni* anabolism were detected, including *lys-C*, *aspS*, *pyrB*, *glnP*, *glnS*, *proA*, *gltB*, *argC*, *hemA*, and *ilvI*. Among the downregulated genes, genes important for *C. jejuni* catabolism were detected, including *htrA*, *pyk*, *Cj0021c*, *Cj1418c*, *Cj1417c*, *purQ*, *aspB*, *Cj0073c*, and *fcl* (Table 1). The network of differentially expressed genes in the *luxS* mutant showed that most of the genes are included in biological processes, such as the tricarboxylic acid (TCA) cycle, the metabolism of pyruvate, nitrogen, and thiamine, and lipopolysaccharide biosynthesis, and these biological processes were upregulated. Many other pathways, including two-component systems such as flagellar assembly, ABC transporters, and protein transport, were also upregulated, whereas many ribosomal genes were downregulated (Fig. 6). Of 30 differentially expressed genes involved in flagellar motility and the colonization of abiotic and biotic surfaces, 22 were upregulated, including *flhF*, *flgI*, *flgB*, *flaG*, *flgM*, *flgD*, *flgG2*, *flgE2*, *fliD*, *flgK*, *fliS*, *flaA*, *pseB*, *pseC*, *ptmA*, *maf4*, *legF*, *Cj1319*, *cj1330*, *cj1026c*, *Cj0391c*, and *Cj0977*. Furthermore, numerous genes important for the stress response were either upregulated (*cstA*, *tpx*, *trxA*, *trxB*, *kata*, *fdxA*, and *ahpC*) or downregulated (*grpE*, *dnaK*, and *htpG*).

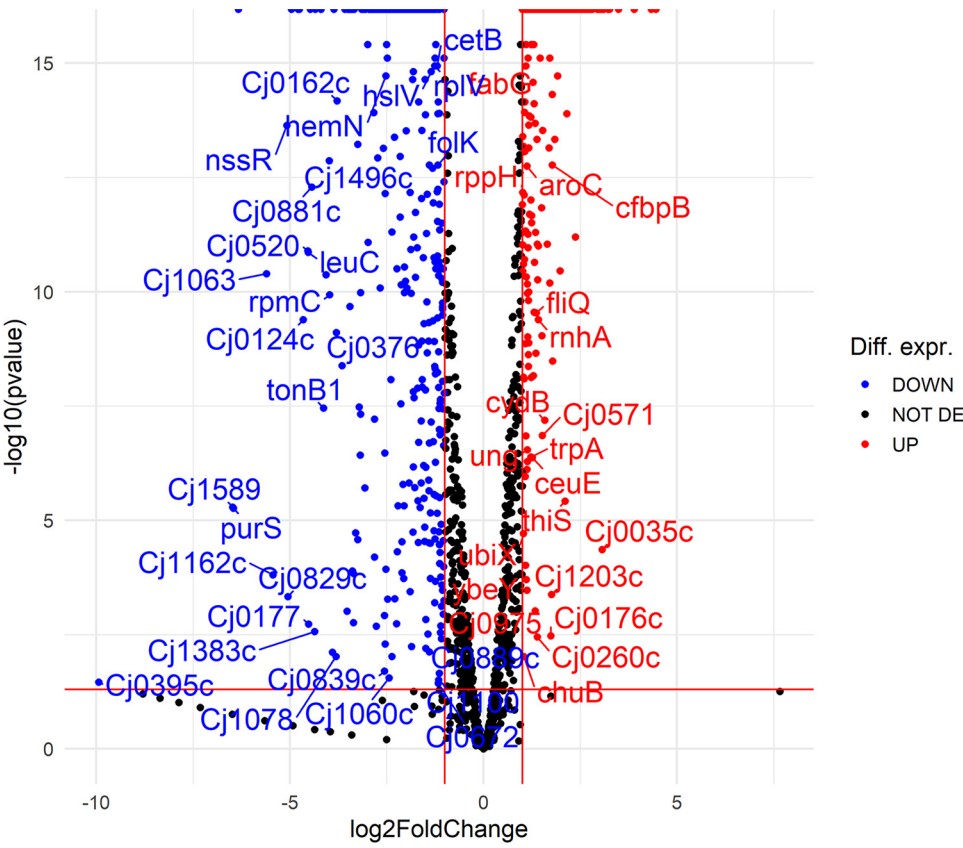

**FIG 5** Volcano plot showing transcriptomic data for *C. jejuni* NCTC 11168 and the *C. jejuni* 11168Δ*luxS* (*luxS* mutant). Blue dots show differentially expressed data for *C. jejuni* NCTC 11168, and red dots show differentially expressed data for the *luxS* mutant. When dots shift more to the left or right, the differential expression (log$_2$ fold change) is stronger. Seven hundred and sixty-five genes were differentially expressed in the *luxS* mutant when normalized to *C. jejuni* NCTC 11168 (FDR, $P \leq 0.05$). Among these, 354 genes were upregulated, and 402 genes were downregulated; Diff. expr., differential expression; DE, differentially expressed.

Further, Gene Ontology (GO) enrichment analysis of the significant differentially expressed genes (FC ≥ 1, $P \leq 0.05$) in the *luxS* mutant compared to the wild type was performed and showed the following results (Fig. 7):

(i)   Biological processes, including oxidation-reduction processes, electron transport chains, and respiratory electron transporter chains, were differentially expressed, but it is not specified whether processes are upregulated or downregulated.

(ii)  Molecular functions were upregulated for secondary active transmembrane transporter activity, symporter activity, solute:cation symporter activity, solute: sodium symporter activity, transporter activity, ion binding, cofactor binding, oxidoreductase activity (acting on NAD[P]H), and oxidoreductase activity (acting on NAD[P]H, quinone, or similar compounds) and were downregulated for rRNA binding.

**Proteomic profile of the *luxS* mutant and wild type.** We further analyzed proteomic profiles to confirm the role of *luxS* in the life cycle of *C. jejuni* by comparing the wild-type and *luxS*-mutant strains. Detailed analysis revealed that a total of 206 proteins were identified and quantified in the samples. Of these proteins, 53 were differentially expressed (FDR, $P \leq 0.05$) (Fig. 8). The largest fold change belonged to *luxS*, as expected, reflecting the effectiveness of the deletion mutation in the mutant strain. Of the other 52 differentially expressed proteins, 32 were upregulated, and 20 were downregulated in the *luxS* mutant compared to the wild type (FDR, $P \leq 0.05$) (Fig. 8).

Specifically, the selected proteins were statistically different in the *luxS* mutant compared with the wild type ($P \leq 0.05$). The upregulated proteins in the *luxS* mutant consisted

**TABLE 1** Differentially expressed genes (FDR, $P \leq 0.05$) that are important for the metabolism of *C. jejuni* 11168Δ*luxS* (*luxS* mutant)

| Gene | log$_2$ fold change | Description |
|---|---|---|
| *lys-C* | 1.13 | Aspartate kinase |
| *aspS* | 1.63 | Aspartate-tRNA ligase |
| *pyrB* | 1.48 | Aspartate carbamoyl transferase catalytic subunit |
| *glnP* | 3.04 | Glutamine transporter |
| *glmS* | 1.57 | Glutamine-fructose-6-phosphate transaminase |
| *proA* | 1.22 | Glutamate-5-semialdehyde dehydrogenase |
| *argC* | 1.76 | *N*-Acetyl-gamma-glutamyl-phosphate reductase |
| *hemA* | 2.33 | Glutamyl-tRNA reductase |
| *ilvI* | 1.66 | Acetolactate synthase I/II/III large subunit |
| *htrA* | −1.60 | Serine protease |
| *pyK* | −2.09 | Pyruvate kinase |
| *Cj0021c* | −1.25 | Fumarylacetoacetate (FAA) hydrolase family protein |
| *Cj1418c* | −1.45 | Glutamine kinase |
| *Cj1417c* | −2.73 | $N^5$-(cytidine 5′-diphosphoramidyl)-L-glutamine hydrolase |
| *purQ* | −2.21 | Glutaminase |
| *aspB* | −1.52 | Aspartate aminotransferase |
| *Cj0073c* | −1.53 | L-Lactate dehydrogenase complex protein LldG |
| *fcl* | −1.60 | GDP-L-fucose synthetase |

of groups with an important role in anabolism (FolE, DapB, Dcd, HemE, ThiG, AroB, PurA, IlvC, GuaA, PurD, DapE, LeuS, HisA, FabH, and PseI), catabolism (GpmI and Eno), cell division (TolB and ParB), and adhesion (PorA). Downregulated proteins played an important role in anabolism (Prs, LegG, LegI, DapA, ThyX, and AcpP), catabolism (Cj1418c), DNA repair (RadA and TopA), cell wall formation and organization (MurB and LpxB), and protein modification (HypA, SelD, and PepA). GO enrichment analysis of the significant differentially expressed proteins in the *luxS* mutant compared to the wild type was performed and showed the following differences in molecular functions (Fig. 9). The upregulated molecular functions consisted mainly of selective molecular interaction with one or more specific sites on another molecule (ATP binding, nucleotide binding, and metal ion binding) and the catalytic activity group. The most important downregulated molecular functions were the metal ion binding group, the transferase activity group, and the DNA binding group.

Networks of differentially expressed proteins (Fig. 10) represent which biological functions are affected in the *luxS* mutant. The upregulated proteins were grouped into annotation clusters with biological functions consisting of the biosynthesis of nucleotides, amino acids, and secondary metabolites, one-carbon metabolism, translation of new proteins, the stress-induced multichaperone system, and cell adhesion. The downregulated biological functions consisted of DNA repair mechanisms, the general stress response, cell wall organization, and biosynthesis and protein modification processes.

## DISCUSSION

Even though *C. jejuni* is a fastidious bacterium with limited growth capabilities outside the host, it is widespread in the food-processing environment and remains the leading cause of foodborne bacterial gastroenteritis in humans worldwide. Little is known regarding how genetic diversity and metabolic capabilities affect their metabolic phenotype and pathogenicity (28). To fill the gap regarding the role of the *luxS* gene in the *C. jejuni* life cycle, we used *C. jejuni* NCTC 11168 (wild type) and *C. jejuni* 11168Δ*luxS* (*luxS* mutant). The mutation in the *luxS* gene can lead to phenotype and transcriptomic changes due to a disrupted methionine cycle and the absence of the AI-2 signaling molecule (14). Thus, in the present study, we evaluated the role of deletion of the *luxS* gene at the physiological, transcriptomic, and proteomic levels under optimal conditions. In this way, we will gain new insights for *C. jejuni* control in the environment by modulation of its *luxS* activity.

Transcriptomic analysis of the *luxS* mutant, which was performed under optimal cultivation conditions (with parameters regarding atmosphere, medium, and temperature

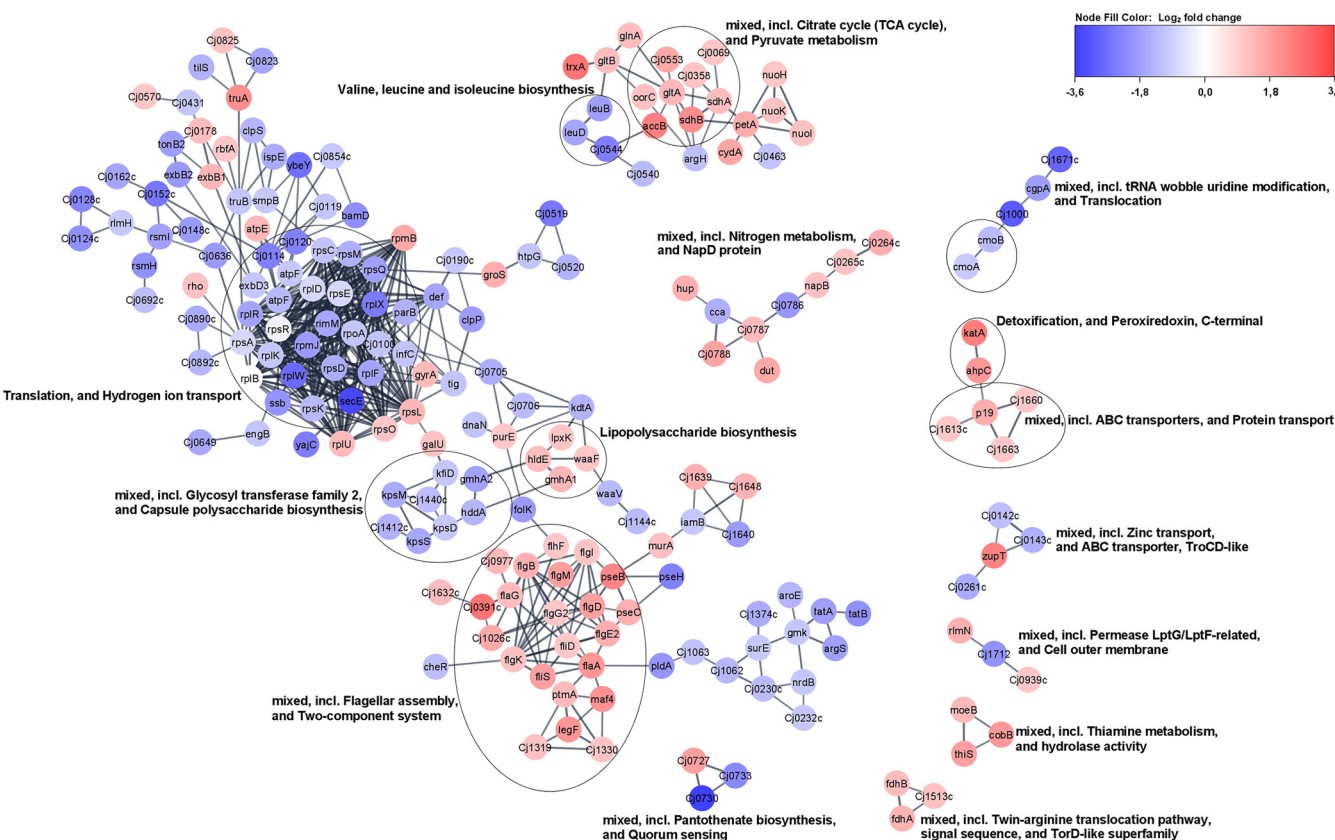

**FIG 6** Network of differentially expressed genes in *C. jejuni* 11168Δ*luxS* (*luxS* mutant). To construct the network of differentially expressed genes, they were searched in a string database for *Campylobacter jejuni* subsp. *jejuni* NCTC 11168 = ATCC 700819 organism with a 0.8 confidence (score) cutoff and a maximum of 10 additional interactors. Afterward, the network was clustered using the MCL Cluster algorithm implemented in the Cytoscape plugin clusterMaker (53) to determine clusters and functional interactions. For each cluster, the built-in functional enrichment was used to obtain enriched terms (available from the stringApp).

according to Davis and DiRita [27]), revealed upregulation of metabolic pathways for the *luxS* mutant when normalized to the wild-type strain. Specifically, when the *luxS* gene is deleted, survival under starvation conditions is better because other metabolic pathways are upregulated, including oxidative phosphorylation, carbon metabolism, citrate cycle, and biosynthesis of secondary metabolites and different essential amino acids (29, 30). Veselovsky et al. (31) stated that during the exponential phase, synthesis of different essential molecules encourages cell growth. The same was determined with proteomic analysis and the Pearson's correlation test, which confirmed a strong correlation between transcriptomic and proteomic data (Fig. S2 and Table S3 in the supplemental material). Those metabolic pathways are necessary for *C. jejuni* energy production and growth (32, 33). Some of the genes that are important for anabolism were upregulated, and some of the genes that are important for catabolism of *C. jejuni* were downregulated in the *luxS* mutant. Upregulation of genes and metabolic pathways included in anabolism can mean that *C. jejuni* synthesizes those essential amino acids itself, so this mutant does not need to take them from its environment. Downregulation of genes included in catabolism of *C. jejuni* is then a logical consequence of upregulation of genes included in the biosynthesis of different essential amino acids. It can be assumed that the *luxS* mutant redirects energy to the upregulation of those other metabolic pathways, which helps the bacteria survive under various conditions. It is interesting to note that Quiñones et al. (21) have shown that the *luxS* mutant has increased chemotaxis to amino acids and decreased chemotaxis to organic acids.

It was also noticed that *cstA*, the gene that encodes carbon starvation protein, was upregulated in the *luxS*-knockout mutant under optimal conditions. It is well known that *cstA* is upregulated under starvation conditions, and it is proposed that CstA has a role in

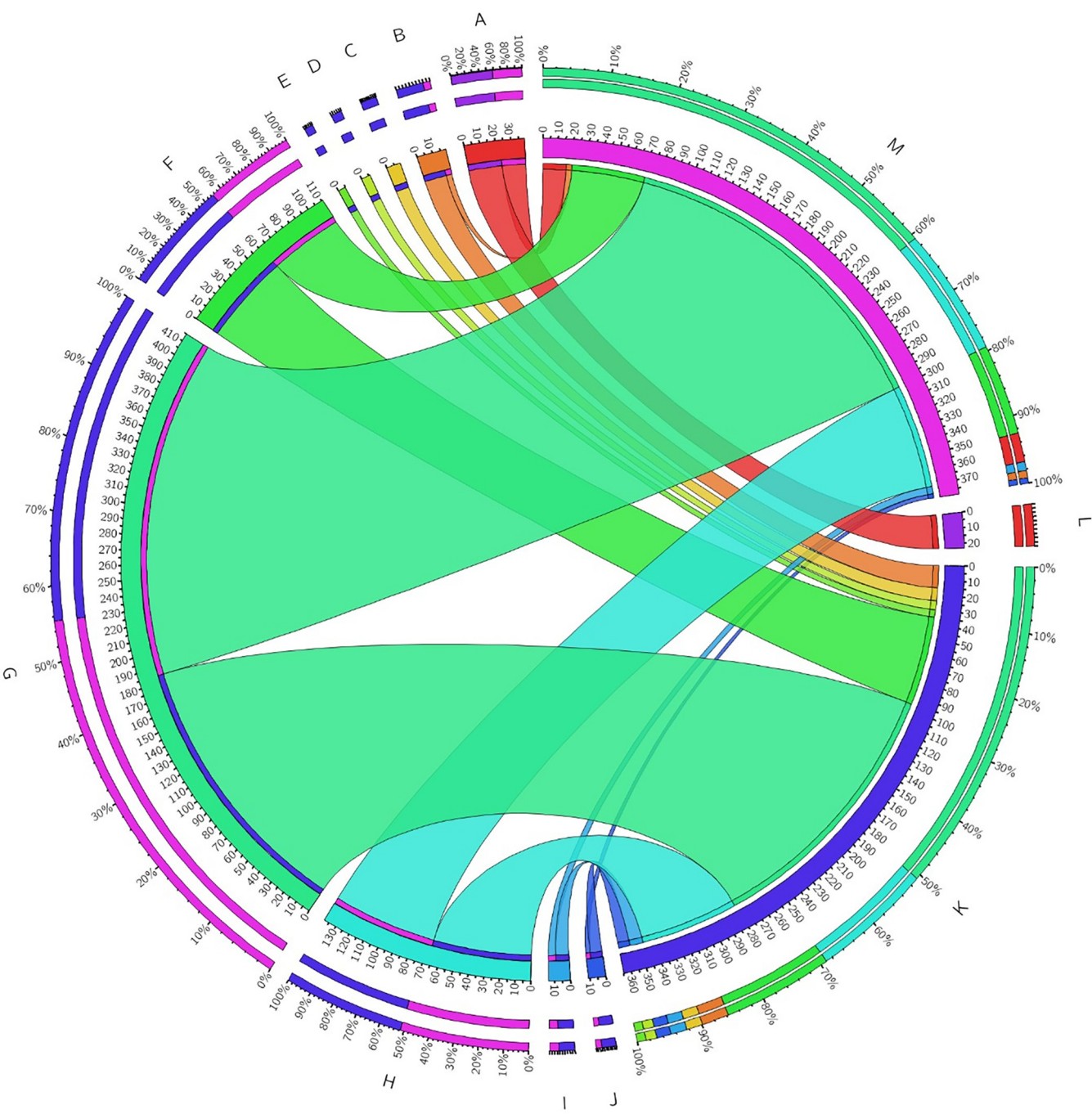

**FIG 7** Functional categorization of molecular functions in *C. jejuni* 11168Δ*luxS* (*luxS* mutant). Blue and pink segments represent the total number of upregulated and downregulated genes, respectively. Functional categories are represented by colored segments, including rRNA binding (A), secondary active transmembrane transporter activity (B), symporter activity (C), solute:cation symporter activity (D), solute:sodium symporter activity (E), transporter activity (F), ion binding (G), cofactor binding (H), oxidoreductase activity (I), and acting on NAD(P)H, oxidoreductase activity, acting on NAD(P)H, quinone, or similar compound as acceptor (J). The number of genes that were upregulated is indicated as K, the number of genes that were downregulated is indicated as L, and no change is indicated as M. To construct the figure, Circos Table Viewer was used.

peptide uptake within *C. jejuni*, which presents an additional source of nutrients necessary for growth (34). Two other genes (*tpx* and *ahpC*) that are important for nutrient starvation (35) were also upregulated in this mutant. Upregulation of these genes is one of the attendant mechanisms that helps this mutant survive. Among the genes that are important for defense against different stresses, *katA*, *trxA*, and *trxB* were upregulated. *katA* encodes catalase in *C. jejuni*, which is important for oxidative defense mechanisms (36). It was also

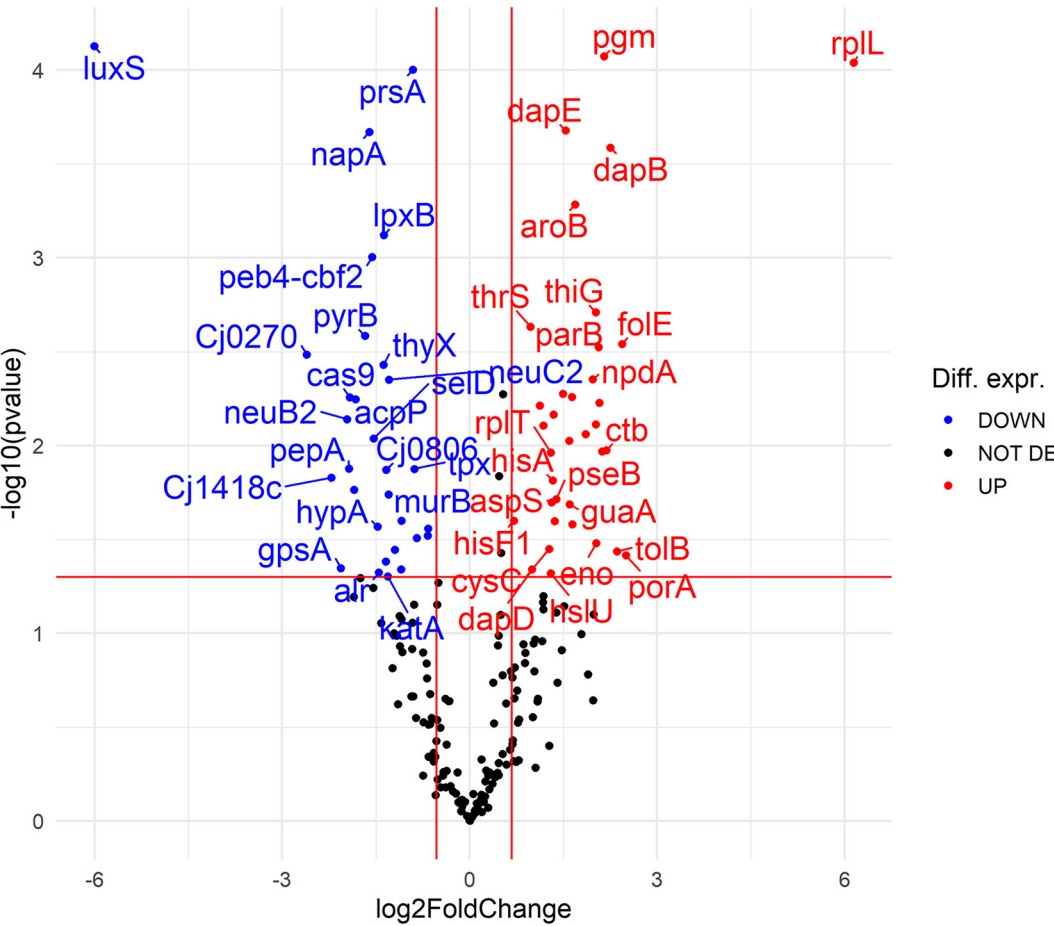

**FIG 8** Volcano plot showing proteomic data for *C. jejuni* 11168Δ*luxS* (*luxS* mutant) compared to wild type. Red dots show proteins with significant fold change in the *luxS* mutant proteome (FDR, $P \leq 0.05$). A left shift indicates downregulation (21 proteins), and a right shift indicates upregulation (32 proteins). The further the dot is from the center, the stronger its differential expression.

shown that *katA* is important for defense against osmotic stress in *C. jejuni* (37). Zhao et al. (38) demonstrated that *C. jejuni* had augmented expression of oxidative stress genes and heat shock genes after exposure to hyperosmotic conditions. Genes *trxA* and *trxB* are also important for antioxidative defense mechanisms in *C. jejuni*, but it was also shown that their expression was increased under acid stress conditions (39). Among the upregulated genes, 18 are important for flagellar motility. Flint et al. (40) noticed that bacterial motility in *C. jejuni* is indirectly connected to resistance to superoxide stress, as mutations of genes involved in flagellar biosynthesis and modification increased *C. jejuni* sensitivity to menadione, a superoxide generator, and slightly to $H_2O_2$. Holmes et al. (24) observed that many genes important for major stress responses were downregulated in the *C. jejuni luxS* mutant, which is contrary to our results. However, these studies were performed under different experimental conditions and time points of cell harvest for transcriptomic analysis. Our results show that the *luxS* mutant has a better adaptive tolerance response (ATR) than the wild type regarding the upregulation of genes important for different stress responses. Different stressful situations activate the general stress response that allows cross-protection for bacteria from various stressors, which means that if cells are exposed to one stressor they also become resistant to another (41). Upregulation of genes important for metabolism occurred in the *luxS* mutant, which indicates that the *luxS* mutant has a wider range of active metabolic pathways than the wild type under the same cultivation conditions, considering the results from the transcriptomic and proteomic analyses. This was also evident from the results obtained at the physiological level, where growth under stress

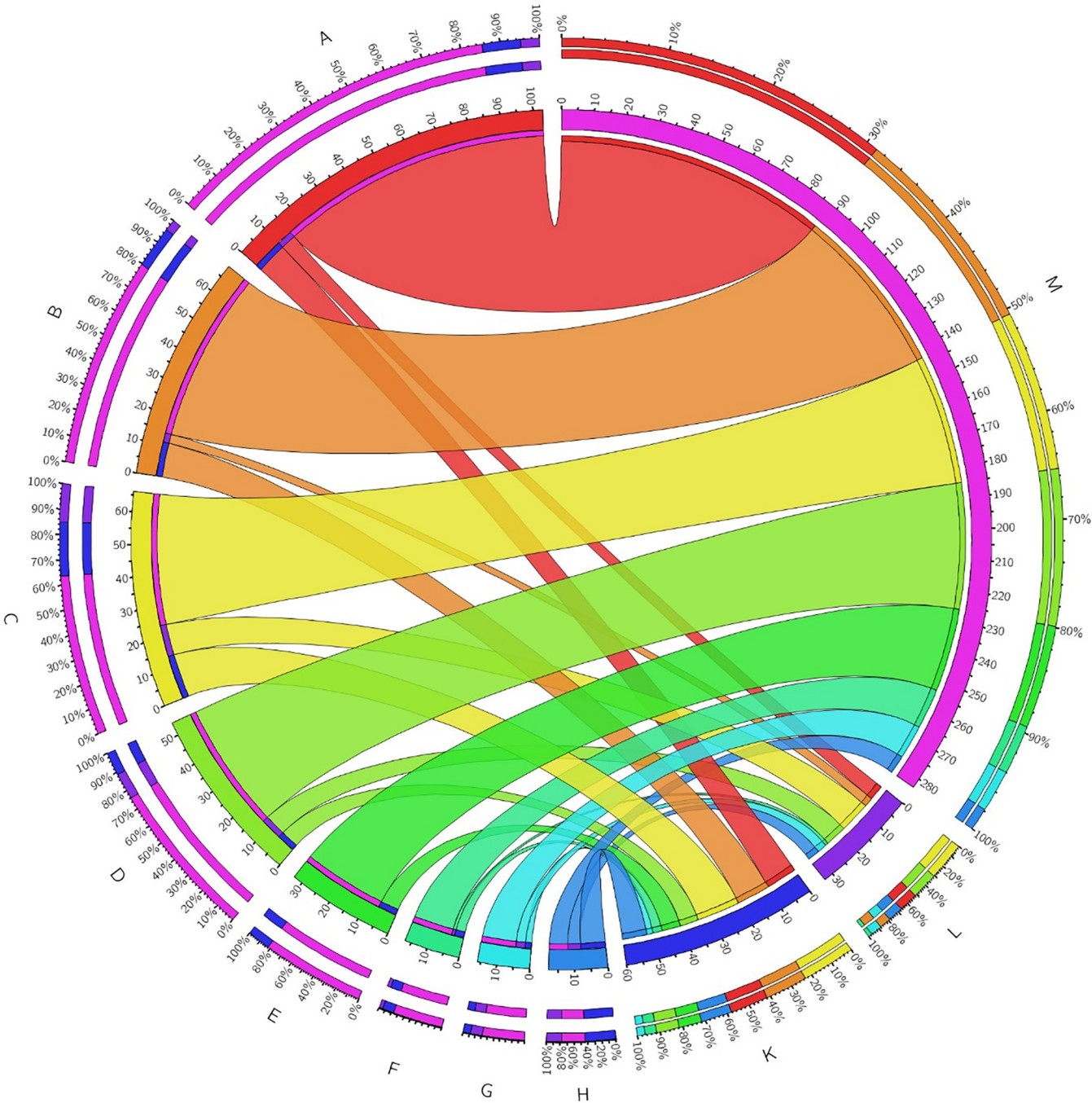

**FIG 9** Functional categorization of molecular functions in the *C. jejuni* 11168Δ*luxS* (*luxS* mutant). Blue and pink segments represent the total number of upregulated and downregulated genes, respectively. Functional categories are represented by colored segments, including ATP binding (A), nucleotide binding (B), metal ion binding (C), transferase activity (D), ligase activity (E), RNA binding (F), DNA binding (G), and catalytic activity (H). The number of genes that were upregulated is indicated as K, the number of genes that were downregulated is indicated as L, and no change is indicated as M. To construct the figure, Circos Table Viewer was used.

conditions (starvation and oxidative stress) and the utilization of different carbon and nitrogen sources were tested. We confirmed that the growth rates of both strains over an extended period of time under optimal conditions were not significantly different to results shown in previous studies (13, 19, 20, 23). These results indicate that the loss of the *luxS* gene is not crucial to the growth and survival of *C. jejuni* over a longer period of incubation under optimal conditions. Nevertheless, differences in survival were observable after the incubation of both strains under stress conditions, which included starvation and oxidative

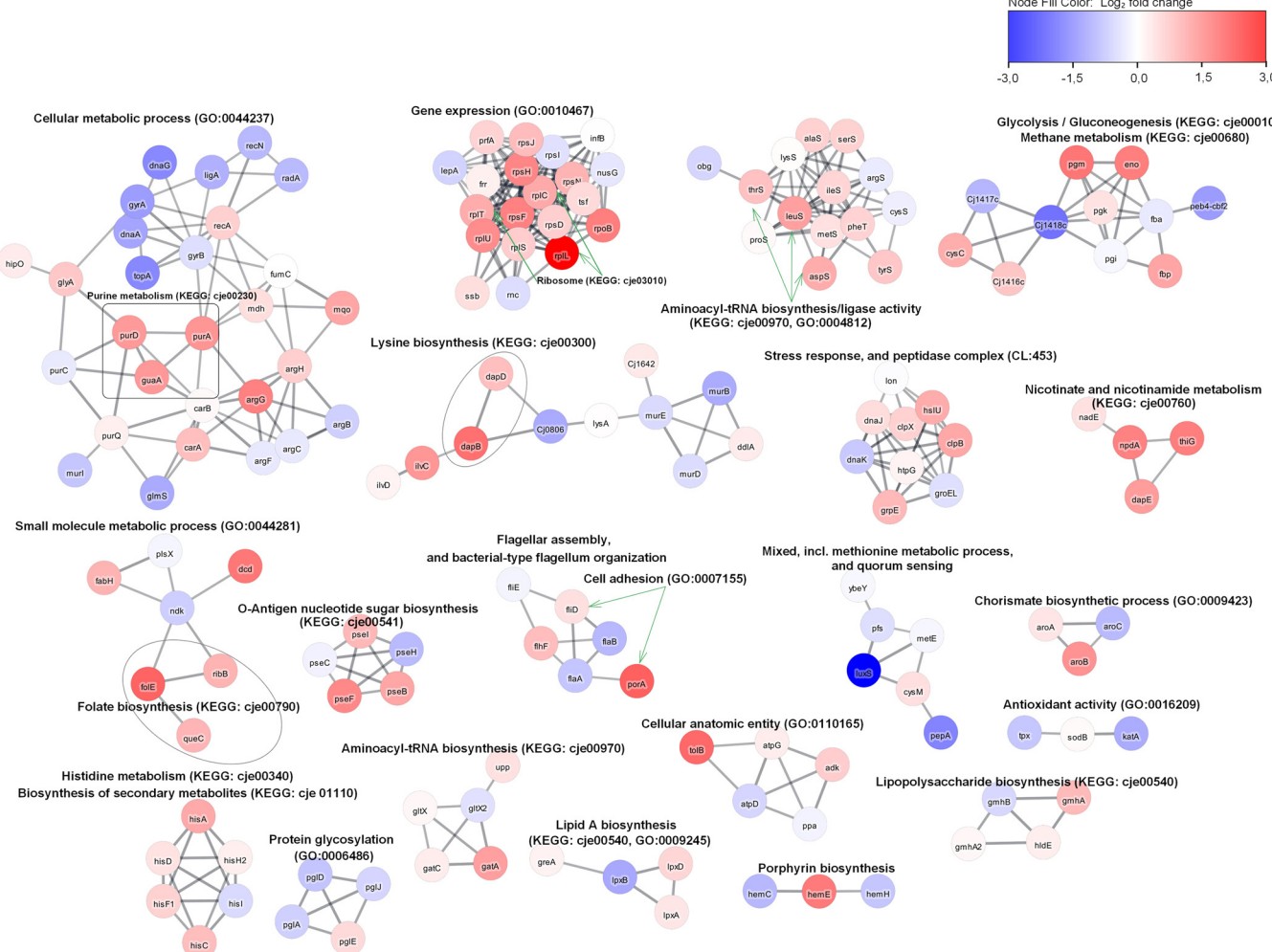

**FIG 10** Network of differentially expressed proteins in *C. jejuni* 11168Δ*luxS* (*luxS* mutant). To construct the network of differentially expressed proteins, they were searched in a string database for *Campylobacter jejuni* subsp. *jejuni* NCTC 11168 = ATCC 700819 organism with a 0.8 confidence (score) cutoff and maximum of 10 additional interactors. Afterward, the network was clustered using the MCL Cluster algorithm implemented in the Cytoscape plugin clusterMaker (53) to determine clusters and functional interactions. For each cluster, the built-in functional enrichment was used to obtain enriched terms (available from the stringApp).

stress. Contrary to expectations of decreased tolerance to environmental stress conditions derived from the lack of an important gene, the survival of the *luxS* mutant was better under low-nutrient conditions than that of the wild type. The results shown in Fig. 2 confirm the results obtained from the transcriptomic analysis, where upregulation of genes important for nutrient starvation were detected even under optimal conditions. These results confirmed that the *luxS* mutant has better ATR than the wild type. For the wild type, it is proposed that incubation in a low-nutrient medium (starvation) triggers VBNC transformation, which prevents colony counting (10). In addition, bacterial shape is an important physiological characteristic and depends mostly on the cell wall, more precisely on the peptidoglycan layer (42). According to growth rates, starvation had a greater effect on the wild type (10) than on the *luxS* mutant; we could not find preserved cells in both strains at the end of the experiment. Analysis of cell morphology with TEM showed that, during starvation, wild-type and *luxS*-mutant cells went from a rod spiral to a coccoid form. In the VBNC state, cells are characterized by a reduction in size and coccoid cellular morphology (43). Kassem et al. (44) reported that several factors induce a VBNC state in *C. jejuni*, including temperature, starvation, formic acid, and aerobic conditions.

Oxidative stress did not significantly influence the survival of either strain, except at the last point (120 h), where the growth of the *luxS* mutant was lower than that of the

wild type. This is in agreement with the results of He et al. (19) who noticed that the *luxS* mutant was more sensitive to oxidative stress than the wild type. However, significant morphological changes in the shape of wild-type and *luxS*-mutant cells, cultivated under aerobic conditions, were noticed after the examination of cells with TEM. Cells went from a rod shape to a coccid form. This is in agreement with the results of Kim et al. (41) who also reported significant morphological changes in *C. jejuni* after exposure to oxygen-rich conditions. Oh et al. (45) demonstrated that increased oxidative stress induces the formation of VBNC in *C. jejuni*. We could conclude that the coccoid form prevailed sooner in the *luxS* mutant than in the wild type.

In addition, we wanted to see if the *luxS* mutant would grow as well as the wild type on 288 different carbon and nitrogen substrates. The Biolog phenotype microarray analysis revealed that the *luxS* mutant poorly utilized only 22 different carbon and nitrogen sources compared to the wild type. The nitrogen sources included L-proline, L-serine, L-glutamine, and L-asparagine, which are the most important growth-promoting amino acids for *C. jejuni* (32). Insights into the transcriptional profile of the *luxS* mutant showed that different metabolic pathways were upregulated, including pathways important for the biosynthesis of different essential amino acids, such as serine, glutamate, valine, leucine, alanine, and arginine (Fig. 7). These results could explain observations at the phenotypic level gained with the Biolog system. Anabolism of the *luxS* mutant is upregulated, and, consequently, the synthesis of essential amino acids is higher, so this mutant does not need to acquire certain amino acids from its environment. The results also showed that the *luxS* mutant poorly metabolized L-fucose, even though it is known that the wild type is capable of metabolizing fucose (32). This has also been observed in other isolates of *C. jejuni* (46). Overall, we can say that the *luxS* mutant had lower growth and survival capacity on these 22 different carbon and nitrogen sources than the wild type.

To conclude, transcriptomic and proteomic analyses revealed major biological differences regarding the TCA cycle, the metabolism of pyruvate, nitrogen, and thiamine, and lipopolysaccharide biosynthesis between the *luxS* mutant and wild type. These processes were upregulated, possibly for the self-preservation of the *luxS* mutant. These novel findings suggest that the *luxS* mutant is better prepared for stressful conditions than the wild type. The *luxS* mutant survived better under starvation conditions and utilized various carbon and nitrogen sources almost as well as the wild type. Moreover, the results presented here significantly complement the recent and very important finding that concentrations of AI-2 signaling molecules (the product of the *luxS* gene) increase linearly with increasing cell concentration (17). Thus, we confirm the strong involvement of *luxS* in the central metabolic pathway, beyond the true intercellular signaling system.

## MATERIALS AND METHODS

**Bacterial strains and growth conditions.** *C. jejuni* NCTC11168 (wild type) and *C. jejuni* 11168Δ*luxS* (*luxS* mutant) (16) were stored in a 800 $\mu$L:200 $\mu$L solution of Mueller-Hinton (MH) broth (Oxoid, UK) and glycerol (Kemika, Croatia) at −80°C. The frozen stock was transferred onto Karmali agar (Oxoid, UK) with a selective supplement (SR0167E, Oxoid, Basingstoke, Hampshire, UK) and incubated for 48 h under microaerobic conditions (5% $O_2$, 10% $CO_2$, 85% $N_2$) at 42°C. A pure culture was transferred to MH agar (Oxoid, UK) without antibiotic (wild type) or supplemented with kanamycin (30 mg/L; Sigma-Aldrich, Germany; *luxS* mutant). Overnight cultures were used in further experiments.

**Determination of growth curves for the *luxS* mutant and wild type.** The cell morphologies of *C. jejuni* NCTC11168 (wild type) and *C. jejuni* 11168ΔluxS (*luxS* mutant) were examined under optimal and stressful conditions (starvation and oxidative stress). Optimal conditions included a microaerobic atmosphere (5% $O_2$, 10% $CO_2$, 85% $N_2$), optimal medium for cultivation (MH medium), and optimal temperature (42°C) (27). Cultures of the *luxS* mutant and wild type were adjusted to an optical density at 600 nm ($OD_{600}$) of 0.1 arbitrary units (AU), which corresponds to (1.1 $\pm$ 0.1) $\times$ $10^7$ CFU/mL (Table S2 in the supplemental material). For growth curve determination, these cultures were then diluted 10,000-fold to reach 5 $\times$ $10^3$ CFU/mL in 5 mL of MH broth and further incubated at 42°C under microaerobic or aerobic conditions for 120 h. Samples were obtained at time points 0, 4, 8, 12, 24, 48, 72, 96, and 120 h. To prepare the starved culture, cells were harvested by centrifugation ($OD_{600}$ = 0.1, 12,000 $\times$ *g*, 4°C, 5 min), washed, resuspended in Ringer solution, as described previously (47), and incubated for 120 h under microaerobic conditions at 42°C. For the aerobically stressed culture, the $OD_{600}$ of 0.1 culture in Mueller-Hinton broth (MHB) was incubated under aerobic conditions at 42°C, and a control incubated under a

microaerobic atmosphere (optimal condition) was used. Samples were obtained at time points 0, 24, and 120 h. All experiments were performed in three or more biological replicates. The concentration of bacterial cells was determined as CFU/mL.

**Morphology assay.** The cell morphologies of *C. jejuni* NCTC11168 (wild type) and *C. jejuni* 11168Δ*luxS* (*luxS* mutant) were followed and assessed with transmission electron microscopy (TEM) under optimal and stressful (starvation and oxidative) conditions as described above. Cell suspensions were applied on Formvar and carbon-coated grids. After 5 min, the sample was soaked away and stained with 1% uranyl acetate. Samples were observed with a Philips (Amsterdam, the Netherlands) CM 100 TEM operating at 80 kV, and images were recorded by an ORIUS SC200 charge-coupled device (CCD) camera using DigitalMicrograph software, Gatan, Inc. (Washington, DC, USA).

**Determination of the utilization of carbon and nitrogen sources with the Biolog system for the *luxS* mutant and wild type.** Cultures of the *luxS* mutant and wild type were prepared in IF-0a inoculating fluid (Biolog, Hayward, CA, USA), mixed with redox indicator Dye D, and dispensed (100 $\mu$L/well) using carbon and nitrogen utilization (PM1, PM2A, and PM3B) phenotypic plates (Biolog) according to the manufacturer's instructions. PM1, PM2A, and PM3B contain 288 different carbon or nitrogen sources. The PM Biolog plates were placed in plastic bags together with microaerobic atmosphere-generating sachets (Oxoid), sealed, and fixed in the automatic plate reader (Omnilog, Biolog, USA) trays using adhesive tape. Plates were incubated at 42°C, and growth on the metabolism substrates in the PM Biolog plates was measured spectrophotometrically after 48 h, as has been described before (48). The cutoff for significance in the relative fold change of growth on phenotypic plates was set at >2.

**Cultivation of the *luxS* mutant and wild type for transcriptomic and proteomic analyses.** The *luxS* mutant and wild type were analyzed after incubation for 16 h at 42°C in a microaerobic atmosphere, and strains were cultivated until the middle of the exponential phase as previously determined by Ramić et al. (12). The cells were harvested by centrifugation (5,000 $\times$ *g*, 5 min, 4°C) and resuspended in 1 mL of cell suspension for further transcriptomic and proteomic analyses.

**Transcriptomic analysis.** Transcriptomic analysis of *C. jejuni* NCTC11168 (wild type) and *C. jejuni* 11168Δ*luxS* (*luxS* mutant) was followed and assessed under optimal conditions (optimal concentration of nutrients and $O_2$).

**(i) RNA isolation and quantification.** Cell lysis for the isolation of total RNA was performed using RNA isolation (TRI) reagent (Sigma-Aldrich, Germany), DNase treatment was performed using PureLink DNase kits (Thermo Fisher Scientific, USA), and purification of isolated RNA was performed using PureLink RNA mini kits (Thermo Fisher Scientific, USA), following the manufacturer's instructions. Quantification and qualification of the total RNA quality was determined using Qubit RNA high-sensitivity assay kits (Thermo Fisher Scientific, USA) and fluorimeter measurements (Qubit v4, Thermo Fisher Scientific, USA). mRNA was enriched from the total RNA using magnetic beads (NEXTflex PolyA, PerkinElmer, USA).

**(ii) Ion Torrent library preparation and sequencing.** For next-generation sequencing (RNA sequencing [RNA-seq]), whole-transcriptome libraries were constructed using the Ion Total RNA-seq kit v2 (Thermo Fisher Scientific, USA). Briefly, mRNA samples were enzymatically fragmented and purified using magnetic beads. Afterward, ion adaptors were hybridized onto the fragmented mRNA and ligated, and reverse transcription was performed. The prepared cDNA samples were purified using magnetic beads, and each cDNA sample was barcoded with an Ion Xpress RNA-seq barcode BC primer (Thermo Fisher Scientific, USA). The cDNA libraries were purified, and the concentration and size distribution of cDNA libraries were determined using an Agilent 2100 Bioanalyzer and an Agilent high-sensitivity DNA kit (Agilent Technologies, USA). The barcoded cDNA libraries that were prepared were diluted to the same molar concentration, pooled in equal volumes, and amplified using an Ion OneTouch 2 system with the accompanying Ion PI Hi-Q OT2 200 kits. Sequencing was performed on an Ion Proton system using Ion PI Hi-Q sequencing 200 kits (all Thermo Fisher Scientific, USA).

**(iii) Data analysis.** The bioinformatics analysis was performed using a CLC Genomics Workbench (version 12.0.3) and a CLC Genomics Server (version 11.0.2; Qiagen, Germany). Before the differential expression analysis, quality control of the sequencing reads and trimming of the adapter sequences were performed using the "Trim Reads" tool. Sequencing reads from each library were subjected to differential expression analysis using the RNA-Seq Analysis 2.21 tool (CLC Genomics Server 20.0.2). The *C. jejuni* subsp. *jejuni* NCTC 11168 (ATCC, 700819) complete genome sequence and genome annotation from the NCBI nucleotide database (accession number: NC_002163.1) was used as the reference genome sequence (49). To compare gene expression between *C. jejuni* NCTC 11168 and *C. jejuni* 11168Δ*luxS*, the "Differential Expression in Two Groups 1.1" tool was used. Genes with absolute $\log_2$ fold change values of $\geq$1 and false discovery rate (FDR) *P* values of $\leq$0.05 were considered differentially expressed. Differentially expressed genes were further analyzed via the STRING Consortium 2020, which provides functional enrichment analysis of protein-protein interaction networks in the STRING mapper tool (https://string-db.org/cgi/input.pl?sessionId=oO8HWWKYl5Fd&input_page_show_search=on).

Gene enrichment analysis was performed using GO_MWU (50), which uses Mann-Whitney *U* tests and the Benjamini-Hochberg FDR corrections of *P* values to define what enriched gene ontology (GO) categories are significantly represented by either upregulated or downregulated genes. GO categories with Benjamini-Hochberg FDR *P* values of <0.1 were considered significantly enriched by either upregulated or downregulated genes.

**Proteomic analysis.** Proteomic analysis of *C. jejuni* NCTC11168 (wild type) and *C. jejuni* 11168Δ*luxS* (*luxS* mutant) was followed and assessed under optimal conditions (optimal concentration of nutrients and $O_2$).

**(i) Protein isolation.** Biomass was resuspended in 300 $\mu$L of lysis buffer consisting of 7 M urea (Sigma-Aldrich, Germany), 2 M thiourea (Sigma-Aldrich, Germany), 5 mM dithiothreitol (DTT; Roche Diagnostics

GmbH, Germany), and 2% (mass/vol) CHAPS (GE Healthcare Life Sciences, Germany). The suspensions were quickly frozen in liquid nitrogen, thawed in warm water, and vortexed. This step was repeated three times. Sterilized zirconia glass beads (Carl Roth, Germany) were then added and homogenized with the Bullet Blender (Next Advance, USA). The homogenized biomass was centrifuged (20,000 × *g*, 20 min, 4°C) to separate cell debris. The supernatant was stored at −80°C for further use.

**(ii) Protein purification.** Protein concentration was measured using the Bradford method-based Bio-Rad protein assay according to the manufacturer's instructions. The calculated volume, consisting of 25 $\mu$g of protein, was transferred to Nanosep microcentrifuges with 30,000 centrifugal Omega filters (Pall, USA). Protein purification followed the protocol of Distler et al. (51) with a slight modification in enzymatic digestion. Lys-C enzyme (New England Biolabs, USA) was used at a ratio of 1:100 (mass of Lys-C enzyme:mass of sample proteins) and incubated for 2 h at 37°C. Then, 50 mM ammonium bicarbonate (Thermo Fisher Scientific, USA) was added to dilute the urea concentration to 1 M, and added trypsin (New England Biolabs, USA) was allowed to work at a ratio of 1:25 (mass of trypsin enzyme:mass of sample proteins) for 16 h at 37°C. To stop proteolysis, we added 20% acetonitrile and 10% formic acid.

Peptides were separated by nano liquid chromatography (Thermo Scientific, Ultimate RSLC 3000) coupled in line with a Q Exactive mass spectrometer equipped with an Easy-Spray source (Thermo Fischer Scientific, USA). Peptides were trapped onto a $C_{18}$ PepMac100 precolumn (300-$\mu$m inner diameter [i.d.] × 5 mm, 100 Å; Thermo Fischer Scientific, USA) using solvent A (0.1% formic acid, high performance liquid chromatography [HPLC] grade water). The peptides were further separated onto an Easy-Spray rapid-separation liquid chromatography (RSLC) $C_{18}$ column (75-$\mu$m i.d., 50-cm length; Thermo Fischer Scientific) using a 60-min linear gradient (15% to 35% solvent B [0.1% formic acid in acetonitrile]) at a flow rate of 200 nL/min. The raw data were acquired on a mass spectrometer in data-dependent acquisition mode. Full-scan mass spectrometry (MS) spectra were acquired on the Orbitrap (scan range of 350 to 1, 500 *m/z*, resolution of 70,000, automatic gain control [AGC] target of 3 × 10⁶, and maximum injection time of 50 ms). The 10 most intense peaks were selected for higher-energy collision dissociation (HCD) fragmentation at 30% normalized collision energy. HCD spectra were acquired on the Orbitrap at a resolution of 17,500, AGC target of 5 × 10⁴, maximum injection time of 120 ms, and a fixed mass of 180 *m/z*. Charge exclusion was selected for unassigned and 1+ ions. The dynamic exclusion was set to 20 s.

**(iii) Database searching.** Tandem mass spectra were searched using SEQUEST HT within proteome discoverer PD1.4 (Thermo Fischer Scientific, version 1.4.0.288) against a database containing 1,632 proteins, of which we used 467 that were manually annotated and reviewed by UniProtKB curator entries, combining sequences from *C. jejuni* (UniProt release from 2021 September 16) and common contaminants. During database searches, cysteines were considered to be fully carbamidomethylated (+57,0215, statically added), methionines were considered to be fully oxidized (+15,9949, dynamically added), and all N-terminal residues and lysines were considered to be acetylated (+42,0106, dynamically added). Two missed cleavages were permitted. Peptide mass tolerance was set at 50 ppm and 0.02 Da on the precursor and fragment ions, respectively. Protein identification was filtered at an FDR below 1%.

**(iv) Quantitative proteomics and statistical analyses.** The quantitative analysis was based on a label-free quantitation method using normalized spectral abundance factor (NSAF) as a measure of relative protein abundance within the protein mixture. Spectral abundance factor (SAF) and NSAF values were calculated, as previously described (52), as the number of spectral counts (PSM) that identify a protein divided by the protein length (L). The PSM/L value represents the SAF, which is then divided by the sum of PSM/L for all proteins in the experiment. NSAF values were calculated after contaminants were removed. For better visualization of the data, NSAF values were multiplied by 100 (NSAF100). The statistical analysis was performed on NSAF values from three biological replicates using *t* tests in Perseus. Proteins significantly changing ($P \leq 0.05$ and 0.5 $\log_2$ fold change) in abundance were represented by a Volcano plot generated in R statistical software (version 4.0.3).

**Statistical analysis.** All of the experiments were performed in triplicate as three or more independent experiments. The data are expressed as means ± standard deviations, with statistical analysis performed in IBM SPSS Statistics 23 (Statsoft Inc., Tulsa, OK, USA). To determine the distribution of the data, a Kolmogorov-Smirnov test of normality was performed, and statistical significances were determined using Mann-Whitney tests for two independent means. A Student's *t* test was used to analyze the growth and survival of the *luxS* mutant compared to the wild type. To evaluate the effects of stress conditions on bacterial growth, Student's *t* tests (paired) were used. Correlation analysis using Pearson's method was performed for significantly differentially expressed results of transcriptomic and proteomic expression data. Statistical analysis was performed with IBM SPSS Statistics 23 (Statsoft Inc., Tulsa, OK, USA). A *P* value of ≤0.05 was considered statistically significant.

**Data availability.** The data for this study can be obtained from the corresponding author.

The data sets generated for this study can be found in the NCBI Sequence Read Archive (SRA) repository under BioProject accession number PRJNA747749 and SRA accession numbers SRR15183209, SRR15183225, SRR15183226, and SRR15183227. The mass spectrometry proteomics data have been deposited to the ProteomeXchange Consortium via the PRIDE partner repository with the data set identifier PXD035008.

## SUPPLEMENTAL MATERIAL

Supplemental material is available online only.

**SUPPLEMENTAL FILE 1**, PDF file, 0.6 MB.

## ACKNOWLEDGMENTS

This study was financed by the Slovenian Research Agency ARRS PhD grant number 51861 (leader D.R.) and projects J4-2542, and P4-0116 (leader S.S.M.) and J4-4548, and J4-3088 (leader A.K.); and Instruct-ULTRA (Grant 731005 to A.K.).

We declare that the research was conducted in the absence of any commercial or financial relationships that could be construed as a potential conflict of interest.

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
