## [Reviewer comments · Microbiology Spectrum]

Microbiology Spectrum

The Role of luxS in *Campylobacter jejuni* Beyond Intercellular Signaling

Dina Ramić, Blaž Jug, Katarina Šimunović, Magda Tušek-Žnidarič, Urban Kunej, Natasa Toplak, Minka Kovač, Marjorie Fournier, Polona Jamnik, Sonja Smole Možina, and Anja Klančnik

Corresponding Author(s): Anja Klančnik, Univerza v Ljubljani

Review Timeline:

Submission Date:	July 5, 2022
Editorial Decision:	August 5, 2022
Revision Received:	September 7, 2022
Editorial Decision:	October 13, 2022
Revision Received:	December 13, 2022
Accepted:	January 7, 2023

Editor: livnat afriat-jurnou

Reviewer(s): Disclosure of reviewer identity is with reference to reviewer comments included in decision letter(s). The following individuals involved in review of your submission have agreed to reveal their identity: Laura Huber (Reviewer #1); Cadi Davies (Reviewer #3)

Transaction Report:

DOI: <https://doi.org/10.1128/spectrum.02572-22>

August 5, 2022

Dr. Anja Klančnik
University of Ljubljana
Ljubljana
Slovenia

Re: Spectrum02572-22 (The Role of luxS in Campylobacter jejuni Beyond Quorum Sensing)

Dear Dr. Anja Klančnik:

Link Not Available

Sincerely,

livnat afriat-jurnou

Journals Department
Reviewer comments:

Reviewer #1 (Comments for the Author):

This is a very well done and important study to understand the role of luxS o the survival of C. jejuni. There are a few important comments below that, if addressed, would strengthen the paper and significance.

Minor comments:

Abstract

Line 25: Delete the "-" before to investigate

Importance:

I am having a little trouble understanding the conclusions of this paper as a matter of importance. The importance section suggests that luxS might be an important gene for *C. jejuni* survival in its environment. But the findings of this project point out that the lack of the luxS gene is what leads to the upregulation of genes that improve survival of *C. jejuni* under starvation (Figure 2B).

Results line 172. Perhaps will be of easier understanding if you add in the beginning of the paragraph: "On luxS mutant, the up-regulates..."

Discussion

Line 92. Delete "which".

Line 100. Misspelled "jejuni"

200. Not just in optimal conditions right? You studied it in many different conditions.

212. Double "them"

213. Delete "that happens because"

217. Cite the author's name and year instead of the citation number.

Line 253. I believe you should start your discussions with the results obtained in Fig2. And the transcriptome analysis is the experiments that confirm what was found in figure 2.

Line 287. Not indented.

Line 292. Again here, I believe that your molecular results confirmed what you found in the survival experiments. This would be more appropriately representing the order you presented results.

Line 293. Replace however with "moreover"

Line 294. "recent and very important finding.."

Line 297. I don't believe you have enough evidence to say "rather than" but you could say "beyond".

Reviewer #2 (Comments for the Author):

This study investigates a luxS knockout in comparison to a wild-type strain of *Campylobacter jejuni*. The authors investigate differences in growth rate and survival, morphological, transcriptional differences, and proteomic differences between these strains. One issue that I have with the paper as it is written currently is that there is no central hypothesis that the authors are testing. Why did they set out to investigate these differences between the luxS and WT strains? Further discussion of the importance of luxS in this organism in the introduction would help. There is vague language throughout, and I feel that a better description of the analyses that the authors conducted must be included in their results section. Furthermore, as noted below: Most transcriptomic analyses in ANY organism under different conditions or a knockout will show differential regulation of "metabolic genes". The authors must further explain why they think that it is noteworthy or significant that they find these genes up or down regulated in their results. Is there a way to convince the readers that these loci are actually important for the different phenotypes seen between wild-type and luxS mutants?

At this point I would suggest a revision including:

-Removal of vague language

-Revision of the results section, including additional statistical analyses, and explanation of why they cherry picked metabolic genes/proteins are being important

-Further details about why they chose to investigate this luxS mutant in the first place, and perhaps the addition of their a priori hypotheses about what they would find in this mutant in the introduction.

Line 1: opening line of abstract is vague, which "biological processes" have not been described in relationship to luxS? Why is important to understand the function of this gene in this organism?

Line 45: Resistance to what? Antibiotics?

Line 45-47: There is no real explanation of why luxS is important and why the researchers are studying it.

Line 57: Change "harvest" to "processing"

Line 61-65: Is antibiotic resistance a problem in this species? There are two conflicting sentences on these lines 1) that there are widespread MDR strains, and 2) there is not a lot of resistance in the population? So which one is it?

Line 67: Revise this sentence. This is vague and I do not understand the connection between metabolic changes in the bacteria and food safety. "Understanding metabolic changes is part of basic knowledge and therefore critical for food safety."

Line 69: What is "chicken juice"?

Line 81: "increment of cell concentration" rephrase, awkward word choice.

Line 84-89: Expand upon what we know about how luxS contributes to the listed phenotypes.

Line 91: Which other biological processes, can you give some examples? Do you have a hypothesis about which processes the gene is involved in regulating?

Line 103: What are optimal conditions for these strains? Perhaps mention "common garden conditions"

Line 102-113: Suggestion, combine these two paragraphs.

Line 119: change could to can, what induced the change?

Line 120: Are you talking about the wild-type or mutant strain here?

Line 126-131: Expand on these results please. Include more information on the differences between the two strains and which nutrient sources either survived significantly better on. Include statistical analyses.

Line 140-143: Why did you chose to list these transcripts? Where they the most strongly up or down regulated? The most significant? Of interest for other reasons?

Line 165: What time point or life cycle was the proteome sampled from?

Line 172-183: Again how were the listed proteins selected? Strongest up or down regulated? Etc. What statical analyses were conducted on the proteome?

Line 192: What is meant by "fragile"?

Line 194-196: Vague.

Starting Line 203: Most transcriptomic analyses in ANY organism under different conditions or a knockout etc will show differential regulation of "metabolic genes". Why do the authors think that it is noteworthy or significant that they find these genes in their analyses? Is there a way to convince the readers that these are actually important for the different phenotypes seen between wild-type and luxS mutants?

Line 242: Not sure what "a better metabolism" means?

Line 275: I am not sure what this means?

"poorly utilized only 22 different carbon and nitrogen sources compared to the wild type"

Line 305: When is it appropriate to use kanamycin in the media. Please calrify.

Figure 6 and 9 are confusing. I do not understand these. Why are different sections of the circle connected?

Question: How many replicates were used for RNA-seq conditions for each strain?

How many replicates were used for proteome conditions for each strain?

Staff Comments:

Preparing Revision Guidelines

- Point-by-point responses to the issues raised by the reviewers in a file named "Response to Reviewers," NOT IN YOUR

COVER LETTER.

- Upload a compare copy of the manuscript (without figures) as a "Marked-Up Manuscript" file.
- Each figure must be uploaded as a separate file, and any multipanel figures must be assembled into one file.
- Manuscript: A .DOC version of the revised manuscript
- Figures: Editable, high-resolution, individual figure files are required at revision, TIFF or EPS files are preferred

Please return the manuscript within 60 days; if you cannot complete the modification within this time period, please contact me. If you do not wish to modify the manuscript and prefer to submit it to another journal, please notify me of your decision immediately so that the manuscript may be formally withdrawn from consideration by Microbiology Spectrum.

Abstract

Line 25: Delete the “-“ before to investigate

Importance:

I am having a little trouble understanding the conclusions of this paper as a matter of importance. The importance section suggests that luxS might be an important gene for *C. jejuni* survival in its environment. But the findings of this project point out that the lack of the luxS gene is what leads to the upregulation of genes that improve survival of *C. jejuni* under starvation (Figure 2B).

Results line 172. Perhaps will be of easier understanding if you add in the beginning of the paragraph: “On luxS mutant, the up-regulates...”

Discussion

Line 92. Delete “which”.

Line 100. Misspelled “jejuni”

200. Not just in optimal conditions right? You studied it in many different conditions.

212. Double “them”

213. Delete “that happens because”

217. Cite the author’s name and year instead of the citation number.

Line 253. I believe you should start your discussions with the results obtained in Fig2. And the transcriptome analysis is the experiments that confirm what was found in figure 2.

Line 287. Not indented.

Line 292. Again here, I believe that your molecular results confirmed what you found in the survival experiments. This would be more appropriately representing the order you presented results.

Line 293. Replace however with “moreover”

Line 294. “recent and very important finding..”

Line 297. I don’t believe you have enough evidence to say “rather than” but you could say “beyond”.

The Role of *luxS* gene in *Campylobacter jejuni* Beyond Intercellular Signaling

Authors' replies to Editor's and Reviewers' comments are bellow. All changes in the manuscript are transmitted via track changes and also colored yellow for better transparency.

Reviewer #1 (Comments for the Author):

This is a very well done and important study to understand the role of *luxS* o the survival of *C. jejuni*. There are a few important comments below that, if addressed, would strengthen the paper and significance.

Minor comments:

Abstract

Line 25: Delete the "-" before to investigate

Our response: Thank you for this observation. We deleted "-" before to investigate.

Importance:

I am having a little trouble understanding the conclusions of this paper as a matter of importance. The importance section suggests that *luxS* might be an important gene for *C. jejuni* survival in its environment. But the findings of this project point out that the lack of the *luxS* gene is what leads to the upregulation of genes that improve survival of *C. jejuni* under starvation (Figure 2B).

Our response: thank you for this suggestion. We agree with the reviewer's comment and have therefore corrected and improved the statement in the importance section as follows: "*Campylobacter jejuni* (*C. jejuni*) is the world's leading foodborne bacterial pathogen of gastrointestinal disease in humans. It is a fastidious but widespread organism and the most frequently reported zoonotic pathogen in the EU since 2005. This leads us to believe that *C. jejuni*, which have limited resistance to stress conditions (starvation and oxidative stress) and growth capacity, must benefit significantly from the *luxS* gene, whose role in the life cycle is well known as it regulates many phenotypes, including intercellular signaling system. Surprisingly, this study confirmed for the first time that the deletion of the *luxS* gene strongly affects the central metabolic pathway of *C. jejuni* which improves its survival showing its role beyond the intercellular signaling system."

Results

Line 172. Perhaps will be of easier understanding if you add in the beginning of the paragraph: "On *luxS* mutant, the up-regulates..."

Our response: Thank you for this comment. We have added the name of strain to improve the clarity of the sentence as follows: "The up-regulated proteins of *luxS* mutant consisted of groups with an important role in anabolism..."

Discussion

Line 192. Delete "which".

Our response: We have deleted which.

Line 100. Misspelled "jejuni"

Our response: We have corrected the misspelled word.

Line 200. Not just in optimal conditions right? You studied it in many different conditions.

Our response: We thank you for this question. We have studied the role of the *luxS* gene at the transcriptome and proteome levels only under optimal conditions. The idea was to test the survival of the *luxS* mutant in stress conditions (starvation and oxidative stress) at the physiological level. Indeed, at the physiological level, we found that the *luxS* mutant survived better in starvation-like conditions and there were no differences between survival in aerobic conditions. These results prompted us to investigate the differences in the transcriptome and proteome profiles of the *luxS* mutant in optimal conditions. The results showed us that many metabolic pathways are upregulated even in optimal conditions, including stress defence genes and proteins, which allow the *luxS* mutant to survive better in all conditions.

We have added the sentence to make the conditions used clearer: “Transcriptomic analysis of the *luxS* mutant, which was performed in optimal cultivation conditions (optimal temperature, atmosphere, and nutrient source), revealed up-regulation of metabolic pathways for *luxS* mutant when normalized to wild type, including oxidative phosphorylation, carbon metabolism, citrate cycle, biosynthesis of secondary metabolites, and biosynthesis of different essential amino acids. ”

Line 212. Double "them"

Our response: We have deleted one “them”.

Line 213. Delete "that happens because"

Our response: We have deleted one “that happens because”.

Line 217. Cite the author's name and year instead of the citation number.

Our response: Thank you for this note. We have added the author’s name and year.

Line 253. I believe you should start your discussions with the results obtained in Fig2. And the transcriptome analysis is the experiments that confirm what was found in figure 2.

Our response: Thank you for this suggestion. We have corrected the sentence as follow: “The results showed in Figure 2 confirm the results obtained with transcriptomic analysis, ...”

Line 287. Not identified.

Our response: Thank you for this observation. We corrected that mistake.

Line 292. Again here, I believe that your molecular results confirmed what you found in the survival experiments. This would be more appropriately representing the order you presented results.

Our response: Thank you for this comment. We agree with it and we corrected the sentence as follow: “The *luxS* mutant survived better under starvation conditions and utilized various carbon and nitrogen sources almost as well as the wild type, which is confirmed by the results obtained at the molecular level.”

Line 293. Replace however with "moreover"

Our response: We have replaced “however” with “moreover”.

Line 294. "recent and very important finding."

Our response: We have corrected this sentence to: “Moreover, the results of this manuscript significantly complement the recent and very important finding that the concentration of AI-2 signaling molecules (the product of the *luxS* gene) increase linearly with the increment of cell concentration (17). ”

Line 297. I don't believe you have enough evidence to say "rather than" but you could say "beyond".

Our response: Thank you for this suggestion. We have replaced “rather than” with “beyond”.

Reviewer #2 (Comments for the Author):

This study investigates a *luxS* knockout in comparison to a wild-type strain of *Campylobacter jejuni*. The authors investigate differences in growth rate and survival, morphological, transcriptional differences, and proteomic differences between these strains. One issue that I have with the paper as it is written currently is that there is no central hypothesis that the authors are testing. Why did they set out to investigate these differences between the *luxS* and WT strains? Further discussion of the importance of *luxS* in this organism in the introduction would help. There is vague language throughout, and I feel that a better description of the analyses that the authors conducted must be included in their results section. Furthermore, as noted below: Most transcriptomic analyses in ANY organism under different conditions or a knockout will show differential regulation of "metabolic genes". The authors must further explain why they think that it is noteworthy or significant that they find these genes up or down regulated in their results. Is there a way to convince the readers that these loci are actually important for the different phenotypes seen between wild-type and *luxS* mutants?

At this point I would suggest a revision including:

-Removal of vague language

-Revision of the results section, including additional statistical analyses, and explanation of why they cherry picked metabolic genes/proteins are being important

-Further details about why they chose to investigate this *luxS* mutant in the first place, and perhaps the addition of their a priori hypotheses about what they would find in this mutant in the introduction.

Line 1: opening line of abstract is vague, which "biological processes" have not been described in relationship to *luxS*? Why is important to understand the function of this gene in this organism?

Our response: Thank you very much for this comment. We agree with the reviewer's comment and are grateful for this comment. We have done our best to explain our observations in the following sentences. We could see at the physiological level that the survival of the *luxS* mutant under starvation-like conditions was better than that of the wild type. Moreover, there were no differences in the survival of both strains under aerobic conditions, except for the last point. These results were contrary to our expectations, as we found in the literature that the deletion of the *luxS* gene affects many phenotypes of the *luxS* mutant, including its survival under stress conditions. It has always been described that the *luxS* knockout mutation leads to a reduction in survival under stress conditions, a reduction in motility, virulence, adhesion, biofilm formation and intercellular signalling (Elvers and Park, 2002; Guerry et al., 2006; Jeon et al., 2003; Quiñones et al., 2009; Plummer 2012; Reeser et al., 2007). Therefore, we thought that the *luxS* gene was critical for *C. jejuni* survival, but from the results obtained, it appeared that this was not the case. This prompted us to compare the gene and protein expression of the *luxS* mutant with that of the wild type to see if there are differences between the biological processes that enhance the survival of the *luxS* mutant under stress conditions. Indeed, we found that many biological processes were up-regulated in the *luxS* mutant, including the synthesis of different essential amino acids such as valine, leucine, isoleucine, thiamine, and arginine. Nitrogen and pyruvate metabolism and the citrate cycle, which are also important for amino acid synthesis and energy production, were also up-regulated. Bacteria require large amounts of nitrogen for the synthesis of all major components of the cell, including amino acids, pyrimidines and purines, NAD and amino sugars (Reitzer, 1996). When we saw that crucial metabolic pathways for the synthesis of essential amino acids were up-regulated in the *luxS* mutant, this explained its better survival under starvation-like conditions. In addition, we saw that genes such as *cstA*, *kata*, *tpx*, *ahpC*, *trxA*, and *trxB*, which are important for defence against starvation and oxidative stress, were up-regulated (Rasmussen et al., 2013; Wright et al., 2009; Grant and Park, 1995; Cameron et al., 2012; Birk et al., 2012; Flint et al., 2014). This result also explains the better survival of the *luxS* mutant

under stress conditions. It was observed that flagellar genes and proteins were up-regulated, which are known to be important for survival under oxidative stress conditions (Kim et al., 2015).

We have added the following information to the sentence: “The full role of the *luxS* gene in biological processes, such as essential amino acids synthesis, nitrogen and pyruvate metabolism and flagellar assembly, of *Campylobacter jejuni* (*C. jejuni*) has not been clearly described.”

We have cited following articles in our response:

Elvers KT, Park SF. 2002. Quorum sensing in *Campylobacter jejuni*: detection of a *luxS* encoded signalling molecule. *Microbiology Reading Engl* 148:1475–1481.

Guerry P, Ewing CP, Schirm M, Lorenzo M, Kelly J, Pattarini D, Majam G, Thibault P, Logan S. 2006. Changes in flagellin glycosylation affect *Campylobacter* autoagglutination and virulence. *Mol Microbiol* 60:299–311.

Jeon B, Itoh K, Misawa N, Ryu S. 2003. Effects of quorum sensing on *flaA* transcription and autoagglutination in *Campylobacter jejuni*. *Microbiol Immunol* 47:833–839.

Plummer PJ. 2012. *LuxS* and quorum-sensing in *Campylobacter*. *Front Cell Infect Microbiol* 2:22.

Quiñones B, Miller WG, Bates AH, Mandrell RE. 2009. Autoinducer-2 production in *Campylobacter jejuni* contributes to chicken colonization. *Appl Environ Microbiol* 75:281–285.

Reitzer, LJ. 1996. Ammonia assimilation and the biosynthesis of glutamine, glutamate, aspartate, asparagine, l-alanine and d-alanine, p. 301–407. In F. C. Neidhardt (ed.), *Escherichia coli* and *Salmonella*: Cellular and Molecular Biology, 2nd ed. ASM Press, Washington, D.C.

Reeser RJ, Medler RT, Billington SJ, Jost BH, Joens LA. 2007. Characterization of *Campylobacter jejuni* biofilms under defined growth conditions. *Appl Environ Microbiol* 73:1908–1913.

Rasmussen JJ, Vegge CS, Frøkiær H, Howlett RM, Krogfelt KA, Kelly DJ, Ingmer H. 2013. *Campylobacter jejuni* carbon starvation protein A (CstA) is involved in peptide utilization, motility and agglutination, and has a role in stimulation of dendritic cells. *J Med Microbiol* 62:1135–1143.

Wright JA, Grant AJ, Hurd D, Harrison M, Guccione EJ, Kelly DJ, Maskell DJ. 2009. Metabolite and transcriptome analysis of *Campylobacter jejuni* *in vitro* growth reveals a stationary-phase physiological switch. *Microbiology Reading Engl* 155:80–94.

Grant KA, Park SF. 1995. Molecular characterization of *kataA* from *Campylobacter jejuni* and generation of a catalase-deficient mutant of *Campylobacter coli* by interspecific allelic exchange. *Microbiology Reading Engl* 141:1369–1376.

Cameron A, Frirdich E, Huynh S, Parker CT, Gaynor EC. 2012. Hyperosmotic stress response of *Campylobacter jejuni*. *J Bacteriol* 194:6116–6130.

Birk T, Wik MT, Lametsch R, Knøchel S. 2012. Acid stress response and protein induction in *Campylobacter jejuni* isolates with different acid tolerance. *BMC Microbiol* 12:1–13.

Flint A, Sun YQ, Butcher J, Stahl M, Huang H, Stintzi A. 2014. Phenotypic screening of a targeted mutant library reveals *Campylobacter jejuni* defenses against oxidative stress. *Infect Immun* 82:2266–2275.

Kim JC, Oh E, Kim J, Jeon B. 2015. Regulation of oxidative stress resistance in *Campylobacter jejuni*, a microaerophilic foodborne pathogen. *Front Microbiol* 6:751.

Line 45: Resistance to what? Antibiotics?

Our response: Thank you for this comment. We have improved our statement as follows: “This led us to believe that *C. jejuni*, which has limited stress tolerance (starvation and oxidative stress conditions) and growth capacity, must benefit significantly from the *luxS* gene, whose role in the life cycle is well

known as it regulates many phenotypes, including motility, biofilm formation, host colonization, virulence, autoagglutination, cellular adherence and invasion, oxidative stress, and chemotaxis.”

Line 45-47: There is no real explanation of why *luxS* is important and why the researchers are studying it.

Our response: Thank you for this observation. We have added additional explanation as indicated: “This led us to believe that *C. jejuni*, which has limited stress tolerance (starvation and oxidative stress conditions) and growth capacity, must benefit significantly from the *luxS* gene, whose role in the life cycle is well known as it regulates many phenotypes, including motility, biofilm formation, host colonization, virulence, autoagglutination, cellular adherence and invasion, oxidative stress, and chemotaxis.”

We thank the reviser for this comment, which shows us that we should perhaps also formulate the title of the manuscript more clearly. Thus, Reviewer’s comments encouraged us to make some additional corrections in the title of the manuscript, which is now entitled: “The role of *luxS* gene in *Campylobacter jejuni* beyond intercellular signaling”.

We have also changed “quorum sensing” into “intercellular signaling” throughout the whole manuscript.

We have modified the first paragraph of the discussion as follow: “*C. jejuni* which show limited hardiness and growth capabilities, remains the leading cause of foodborne bacterial gastroenteritis in humans worldwide. Little is known regarding how genetic diversity and metabolic capabilities affect their metabolic phenotype and pathogenicity (27). To fill the gap regarding the role of the *luxS* gene in the *C. jejuni* life cycle, we used *C. jejuni* NCTC 11168 (wild type) and *C. jejuni* 11168 Δ *luxS* (*luxS* mutant). The mutation in the *luxS* gene can lead to phenotype and transcriptome changes due to a disrupted methionine cycle and the absence of the AI-2 signaling molecule (14). Thus, in the present study we evaluated the role of the deletion of the *luxS* gene at physiological level, as well as at transcriptome and proteome level in optimal conditions. In this way we will gain new insights for *C. jejuni* control in the environment by modulation of its *luxS* activity.”

Line 57: Change "harvest" to "processing"

Our response: Thank you for this suggestion. We changed “harvest” to “processing”.

Line 61-65: Is antibiotic resistance a problem in this species? There are two conflicting sentences on these lines 1) that there are widespread MDR strains, and 2) there is not a lot of resistance in the population? So which one is it?

Our response: We thank you for this observation. Yes, *C. jejuni* is highly resistant to several antibiotics, so antibiotic resistance is a major problem for this species (Rozman et al., 2018). We have corrected the misspelled sentence as follows: “Moreover, the “*Campylobacter* paradox” is known worldwide (8), meaning that even though *Campylobacter* is very susceptible bacteria, it is still presented in the food industry and it is the leading cause of human gastrointestinal diseases worldwide.”

We have used following cite in the answer:

Rozman V, Matijašić BB, Smole Možina S. 2018. Antimicrobial resistance of common zoonotic bacteria in the food chain: An emerging threat. *Antimicrob Resist - A Glob Threat*, <https://doi.org/10.5772/INTECHOPEN.80782>.

Line 67: Revise this sentence. This is vague and I do not understand the connection between metabolic changes in the bacteria and food safety. "Understanding metabolic changes is part of basic knowledge and therefore critical for food safety."

Our response: Thank you for this highlight. *Campylobacter jejuni* is a food-borne bacteria which is transmitted to humans through the consumption of contaminated food (usually undercooked, contaminated poultry meat), so *C. jejuni* is a major concern for the food industry and safety. It is necessary to understand the whole life cycle of *C. jejuni*, including its metabolism, in order to find the most appropriate method to help fight against this pathogen bacteria.

We have improved the sentence as follows: "Understanding metabolic changes is part of the basic knowledge about the life cycle of the *C. jejuni* and will help in the fight against this leading food-borne pathogen, ensuring better safety of food products."

Line 69: What is "chicken juice"?

Our response: Thank you for this question, which reminds us that this food model may not be so familiar and known to everyone, and it was our mistake for not writing this more clearly. Poultry products (in various forms/concentrations) are often used for various studies of pathogenic bacteria, especially to study their growth, survival, and biofilm formation. *Campylobacter*, along with *Salmonella enterica*, *Listeria monocytogenes*, and *Escherichia coli*, are among the microbes that pose a significant food safety risk on chicken. However, chicken/meat "juice" can be used as a food-based model system to study the survivability of these microbes. Birk et al. (2004) described it as juice collected from frozen chickens and subsequently cleared by centrifugation and subjected to sterile filtration. Subsequently, several studies have been conducted to investigate the antimicrobial and antibiofilm effects of agents against *Salmonella* Enteritidis, *Listeria monocytogenes*, and *Escherichia coli* in chicken juice (Lee et al., 2021).

Since *Campylobacter* is a natural commensal of poultry, several researchers use chicken juice as a food model for *Campylobacter* habitat (Birk et al. 2004, Piskernik et al., 2011, Klančnik et al., 2014). In addition, organic material residues such as food particles and chicken juice can act as a conditioning layer on surfaces, leading to increased bacterial adhesion (Arnold and Bailey, 2000; Brown et al., 2014; Li et al., 2017; Melo et al., 2017). Ligowska et al. (2011) also reported that *luxS* gene expression was increased in *C. jejuni* cultured in chilled poultry meat juice.

We have improved the clarity of the text as indicated: "Since the gene *luxS* is reported to be up-regulated in surviving cells at low temperatures in chicken juice, a common food model for *C. jejuni*'s natural habitat, and its absence leads to reduced colonization and biofilm formation (11, 12) in this study we focused on evaluating its involvement in the *C. jejuni* life cycle."

We have cited following articles in our response:

Birk T, Ingmer H, Andersen MT, Jørgensen K, Brøndsted L 2003. Chicken juice, a food-based model system suitable to study survival of *Campylobacter jejuni*. *Lett Appl Microbiol* 38,66–71, doi:10.1046/j.1472-765X.2003.01446.x.

Piskernik S, Klančnik A, Tandrup Riedel C, Brøndsted L, Smole Možina S. 2011. Reduction of *Campylobacter jejuni* by natural antimicrobials in chicken meat-related conditions. *Food control*. 22,718-724. <https://doi.org/10.1016/j.foodcont.2010.11.002>.

Klančnik A, Piskernik S, Bucar F, Vučković D, Smole Možina S, Jeršek B. 2014. Reduction of microbiological risk in minced meat by a combination of natural antimicrobials. *J Sci Food Agric* 94: 2758–2765. <https://doi.org/10.1002/jsfa.6621>.

Arnold JW, Bailey GW. Surface finishes on stainless steel reduce bacterial attachment and early biofilm formation: Scanning electron and atomic force microscopy study. *Poult Sci* 2000;79:1839–1845. doi: 10.1093/ps/79.12.1839.

Brown H.L., Reuter M., Salt L.J., Cross K.L., Betts R.P., van Vliet A.H.M. Chicken juice enhances surface attachment and biofilm formation of *Campylobacter jejuni*. *Appl Environ Microbiol.* 2014;80:7053–7060. doi: 10.1128/AEM.02614-14.

Li J., Feng J., Ma L., de la Fuente Núñez C., Gözl G., Lu X. Effects of meat juice on biofilm formation of *Campylobacter* and *Salmonella*. *Int J Food Microbiol.* 2017;253:20–28. doi: 10.1016/j.ijfoodmicro.2017.04.013.

Melo RT, Mendonça E.P., Monteiro G.P., Siqueira M.C., Pereira C.B., Peres P.A.B.M., Fernandez H., Rossi DA. Intrinsic and extrinsic aspects on *Campylobacter jejuni* biofilms. *Front Microbiol* 2017;8:1332. doi: 10.3389/fmicb.2017.01332.

Ligowska M, Cohn MT, Stabler RA, Wren BW, Brondsted L. 2011. Effect of chicken meat environment on gene expression of *Campylobacter jejuni* and its relevance to survival in food. *Int J Food Microbiol.* 145, S111–S115. doi: 10.1016/j.ijfoodmicro.2010.08.027.

Lee DU, Park YJ, Yu, HH, Jung SC, Park JH, Lee DH, Lee NK, Paik HD. Antimicrobial and Antibiofilm Effect of ϵ -Polylysine against *Salmonella* Enteritidis, *Listeria monocytogenes*, and *Escherichia coli* in Tryptic Soy Broth and Chicken Juice. *Foods* 2021, 10, 2211. <https://doi.org/10.3390/foods10092211>.

Line 81: "increment of cell concentration" rephrase, awkward word choice.

Our response: Thank you for this suggestion. We have rephrased this sentence as follow: “We have previously shown that the concentration of AI-2 in *C. jejuni* increases linearly with the increasing of cell concentration, which suggests that this signaling molecule is only a by-product of methyl cycle (17).”

Line 84-89: Expand upon what we know about how luxS contributes to the listed phenotypes.

Our response: Thank you for this comment. We have expanded our explanation as follows: “It has been shown that the intercellular signaling is completely absent in the *luxS* mutant, while motility, biofilm formation, autoagglutination, host colonization and invasion of the *luxS* mutant are decreased in comparison to the wild type (13, 16, 20–26).”

Line 91: Which other biological processes, can you give some examples? Do you have a hypothesis about which processes the gene is involved in regulating?

Our response: Thank you for this question. The *luxS* gene is important for *C. jejuni* central metabolism as well as for different phenotypes, including intercellular signaling, adhesion, biofilm formation, virulence, autoagglutination, host colonization and invasion. Those processes are improved when the *luxS* gene is presented. If the *luxS* gene is deleted than the intercellular signaling is completely absent, while motility, biofilm formation, autoagglutination, host colonization and invasion of the *luxS* mutant are decreased in comparison to the wild type (Elvers and Park, 2002; Guerry et al., 2006; Jeon et al., 2003; Reeser et al., 2007; Quiñones et al., 2009; Plummer 2012). Moreover, the primary role of the *luxS* gene in the *s*-adenosyl-homocysteine recycling pathway, where it is responsible for the hydrolysis of *s*-adenosylhomocysteine to homocysteine, which is then further metabolized to *s*-adenosylmethionine (SAM). The SAM pathway is included in methyl recycling in bacteria, which is associated with bacterial DNA methylation, chemotaxis, motility, and different metabolic and biosynthetic reactions. It is also crucial for bacterial polyamine formation and vitamin synthesis. Alteration of the SAM pathway associated with *luxS* mutagenesis can have significant impacts on bacterial metabolism (Adler et al., 2014). Our hypothesis was that the deletion of the *luxS* gene could significantly influence *C. jejuni*'s metabolism, which will than consequently reduce its survival under stress conditions. According to our results, we were wrong. In *luxS* mutant up-regulation of different metabolic pathways was observed, including up-regulation of the genes important for synthesis of different essential amino acids, such as

valine, leucine, isoleucine, thiamine, arginine. Nitrogen and pyruvate metabolism as well as citrate cycle were also up-regulated, which we believe helped *luxS* mutant to better survive in the starvation like conditions.

We have added some examples of biological processes in the sentence as follow: “Despite it being clear that the *luxS* gene is important for many phenotypes of *C. jejuni*, the role of *luxS* in other biological processes, such as synthesis of different essential amino acids, nitrogen and pyruvate metabolism has not been clearly described to date.”

We have cited following articles in the response:

Elvers KT, Park SF. 2002. Quorum sensing in *Campylobacter jejuni*: detection of a *luxS* encoded signalling molecule. *Microbiology Reading Engl* 148:1475–1481.

Guerry P, Ewing CP, Schirm M, Lorenzo M, Kelly J, Pattarini D, Majam G, Thibault P, Logan S. 2006. Changes in flagellin glycosylation affect *Campylobacter* autoagglutination and virulence. *Mol Microbiol* 60:299–311.

Jeon B, Itoh K, Misawa N, Ryu S. 2003. Effects of quorum sensing on *flaA* transcription and autoagglutination in *Campylobacter jejuni*. *Microbiol Immunol* 47:833–839.

Plummer PJ. 2012. *LuxS* and quorum-sensing in *Campylobacter*. *Front Cell Infect Microbiol* 2:22.

Quiñones B, Miller WG, Bates AH, Mandrell RE. 2009. Autoinducer-2 production in *Campylobacter jejuni* contributes to chicken colonization. *Appl Environ Microbiol* 75:281–285.

Reeser RJ, Medler RT, Billington SJ, Jost BH, Joens LA. 2007. Characterization of *Campylobacter jejuni* biofilms under defined growth conditions. *Appl Environ Microbiol* 73:1908–1913.

Adler L, Alter T, Sharbati S, Gözl G. 2014. Phenotypes of *Campylobacter jejuni luxS* mutants are depending on strain background, kind of mutation and experimental conditions. *PLOS ONE* 9:e104399.

Line 103: What are optimal conditions for these strains? Perhaps mention "common garden conditions"

Our response: Thank you for this question. One of the optimal mediums for *C. jejuni* cultivation is Mueller Hinton (MH) and is composed of dehydrated infusion from beef (300 g/L), casein hydrolysate (17.5 g/L) and starch (1.5 g/L). MH media has the highest recovery rate and is recommended for *C. jejuni* (Davis and DiRita, 2008; Ng, 1985).

We have added explanation for optimal conditions of cultivation as follow: “The *luxS* mutant and the wild type were cultivated in optimal conditions (optimal atmosphere, medium and temperature) to determine basic differences that might arise during their growth.”

With the aim of providing clearer information to readers, we have also improved the text in the "Material and Methods" section. To better describe the optimal cultivation conditions, we added an explanation as follows: “**Determination of growth curves for *luxS* mutant and wild type.** The cell morphology of *C. jejuni* NCTC11168 (wild type) and *C. jejuni* 11168Δ*luxS* (*luxS* mutant) was examined under optimal and stress conditions (starvation and oxidative). Optimal conditions included a microaerobic atmosphere (5% O₂, 10% CO₂, 85% N₂), optimal medium for cultivation (MH medium) and optimal temperature (42 °C).”

References which we cited in our response:

Davis L, DiRita V. 2008. Growth and laboratory maintenance of *Campylobacter jejuni*. *Curr Protoc Microbiol*. Doi: 10.1002/9780471729259.mc08a01s10

Ng LK. 1985. Comparison of basal media for culturing *Campylobacter jejuni* and *Campylobacter coli*. *J Clin Microbiol* 21:226-230.

Line 102-113: Suggestion, combine these two paragraphs.

Our response: We agree with this suggestion and have combined these two paragraphs as follows: “The *luxS* mutant and the wild type were cultivated in optimal conditions (optimal atmosphere, medium and temperature) to determine basic differences that might arise during their growth. Further, the survival of the *luxS* mutant and of the wild type were studied also in starvation (low amount of nutrients) and oxidative stress (aerobic atmosphere) conditions. There were no significant differences in the survival of either strain under optimal conditions during the whole period of cultivation ($P < 0.05$) (Fig. S1 and Fig. 2A). The wild type was more sensitive to starvation ($P < 0.05$) compared to the *luxS* mutant at each time point (Fig. 2B). These differences were more than 3 log₁₀CFU/mL. Oxidative stress had no significant effect on the survival of either strain, except after 120 h, where the survival of wild type was 1 log₁₀CFU/mL higher compared to the *luxS* mutant ($P < 0.05$) (Fig. 2C).”

Line 119: change could to can, what induced the change?

Our response: Thank you for this suggestion. We have changed “could” to “can”. The change was induced by the exposure to stress conditions (starvation and oxidative stress), which is described in more detail in the Discussion section (the third and fourth paragraphs) as follows: “Analysis of cell morphology with TEM showed that, during starvation, cells of the wild type and of the *luxS* mutant went from a rod spiral to a coccoid form. In VBNC state, cells are characterized by a reduction in size and coccoid cellular morphology (40). Kassem et al. (41) reported that several factors induce a VBNC state in *C. jejuni*, including temperature, starvation, formic acid, and aerobic conditions.

Oxidative stress did not significantly influence the survival of either strain, except at the last point (120 h), where the growth of the *luxS* mutant was lower in comparison to the wild type. This is in accordance from the results of He et al. (19) who noticed that the *luxS* mutant was more sensitive to oxidative stress than the wild type. However, significant morphological changes in the cell shape of the wild type and *luxS* mutant, cultivated under aerobic conditions, were noticed after the examination of cells with TEM. Cells went from a rod shape to a coccid form.”

Line 120: Are you talking about the wild-type or mutant strain here?

Our response: Thank you for this question. Here, we are talking about both strains. Both strains (*luxS* mutant and wild type) had helical and rod shaped bacteria, but in the *luxS* mutant rod shape prevailed and the cells were smaller (1.96 μm × 0.67 μm) in comparison with the wild type (2.77 μm × 0.61 μm). We have clarified the sentence as follows: “Both shapes were present in both strains, but in the *luxS* mutant rod shape prevailed and the cells were smaller (1.96 μm × 0.67 μm) in comparison with the wild type (2.77 μm × 0.61 μm).”

Line 126-131: Expand on these results please. Include more information on the differences between the two strains and which nutrient sources either survived significantly better on. Include statistical analyses.

Our response: Thank you for this note which refers to results obtained with the Biolog system. In order to better clarify and expand on these results, we added additional information as follows: “**Utilization of different carbon and nitrogen sources by the *luxS* mutant and the wild type.** Results obtained with the Biolog system showed that the *luxS* mutant poorly metabolized only 22 of 288 different carbon and nitrogen sources compared to the wild type ($P < 0.05$) (Fig. 4). The signal collected from the wild type was >2 times higher than the *luxS* mutant when exposed to oxalomalic acid, pyruvic acid, glucuronamide and D-psicose, and >10 times higher when exposed to methyl-pyruvate, mono methyl succinate, L-proline, L-serine, fumaric acid, citric acid, butyric acid, keto-glutaric acid, D and L-malic acid, glutamic acid, lactic acid, aspartic acid, succinic acid, L-glutamine, L-asparagine, and L-fucose,

implying that the metabolism of the *luxS* mutant was limited upon exposure to these nutrients ($P < 0.05$). Furthermore, this means that the growth and survival ability of the *luxS* mutant on these sources is lower than that of the wild type, which grew better than the *luxS* mutant on these 22 sources.”

Line 140-143: Why did you chose to list these transcripts? Where they the most strongly up or down regulated? The most significant? Of interest for other reasons?

Our response: Thank you for this question, which indicates that the text has not been adequately described and thus the reader would miss important information. However, the list of these transcripts was selected because they were statistically different in the *luxS* mutant compared with its wild type ($P < 0.05$). Some of these genes were selected because they explain the observations we made at the physiological level. For example, some genes important for anabolism (*lys-C*, *aspS*, *pyrB*, *glnP*, *glnS*, *proA*, *gltB*, *argC*, *hemA*, and *ilvI*) were up-regulated, as were genes for stress defence and flagellar assembly, and we could see that the *luxS* mutant survived better in the starvation conditions. Moreover, the *luxS* mutant poorly utilized only 22 out of 288 different carbon and nitrogen sources compared to the wild type ($P < 0.05$), and the explanation for this may lie in the downregulation of some genes important for catabolism (*htrA*, *pyk*, *Cj0021c*, *Cj1418c*, *Cj1417c*, *purQ*, *aspB*, *Cj0073c*, and *fcl*) of *C. jejuni*.

We have clarified the paragraph as follows: “Detailed analysis showed that 765 genes were differentially expressed in the *luxS* mutant when normalized to the wild type (FDR, $P \leq 0.05$). Among them, 354 genes were up-regulated and 402 genes were down-regulated (Supplementary Table 1; Fig. 5). However, the list of these transcripts was selected because they were statistically different in the *luxS* mutant compared with its wild type ($P < 0.05$). Some of these genes were selected because they explain the observations we made at the physiological level. Among the up-regulated genes, genes important for *C. jejuni* anabolism were detected, including *lys-C*, *aspS*, *pyrB*, *glnP*, *glnS*, *proA*, *gltB*, *argC*, *hemA*, and *ilvI*. Among the down-regulated genes, genes important for *C. jejuni* catabolism were detected, including *htrA*, *pyk*, *Cj0021c*, *Cj1418c*, *Cj1417c*, *purQ*, *aspB*, *Cj0073c*, and *fcl* (Table 1). The network of differentially expressed genes in the *luxS* mutant showed that most of the genes are included in biological processes, such as TCA cycle, pyruvate, nitrogen, and thiamine metabolism, as well as in lipopolysaccharide biosynthesis, and that these biological processes were up-regulated. Many other pathways, including the two-component system, were also up-regulated, while many ribosomal genes were down-regulated (Fig. 6). Of 30 differentially expressed genes involved in flagellar motility and the colonization of abiotic and biotic surfaces, 22 were up-regulated: *flhF*, *flgI*, *flgB*, *flaG*, *flgM*, *flgD*, *flgG2*, *flgE2*, *fliD*, *flgK*, *fliS*, *flaA*, *pseB*, *pseC*, *ptmA*, *maf4*, *legF*, *Cj1319*, *cj1330*, *cj1026c*, *Cj0391c*, and *Cj0977*. It was also noticed that different genes important for stress response were up-regulated, including *cstA*, *tpx*, *trxA*, *trxB*, *kata*, *fdxA*, and *ahpC*, but some of the genes important for general stress response were down-regulated, i.e. *grpE*, *dnaK*, and *htpG*.”

Line 165: What time point or life cycle was the proteome sampled from?

Our response: Thank you for this question. Samples for proteome and transcriptome analysis of the *luxS* mutant and wild type were collected after 16 h incubation at 42°C in a microaerobic atmosphere. This time point represents middle of the exponential phase of *C. jejuni* growth curve, which was previously published by Ramić et al (2021).

We have clarified the paragraph in the "Materials and Methods" section as follows: “**Cultivation of *luxS* mutant and wild type for transcriptomic and proteomic analysis.** The *luxS* mutant and wild type were analyzed after incubation for 16 h at 42 °C in a microaerobic atmosphere, and thus strains were cultivated until the middle of the exponential phase as previously determined by Ramić et al. (12). The cells were harvested by centrifugation (5,000 × g, 5 min, 4°C) and resuspended to 1 mL cell suspension for further transcriptomic and proteomic analysis.”

Line 172-183: Again how were the listed proteins selected? Strongest up or down regulated? Etc. What statistical analyses were conducted on the proteome?

Our response: We thank you for this question, which we have clarified in the manuscript. The selected and listed proteins were statistically different in the *luxS* mutant compared with the wild type ($P < 0.05$; \log_2 fold-change > 0.5). A database of manually annotated proteins reviewed by UniProtKB curators was created. All indicated proteins were up- or down-regulated and grouped according to their molecular and biological functions to facilitate interpretation and comparison with other results. This is described in detail in the Materials and Methods sections in **(iii) Database searching**. On your suggestion we have clarified this also in Results section in the manuscript with the sentence as follows: Specifically, the selected proteins were statistically different in the *luxS* mutant compared with the wild type ($P < 0.05$).

The statistical analysis was performed on NSAF values from three biological replicates using paired t-test in Perseus. Proteins significantly changing ($P < 0.05$ and $0.5 \log_2$ fold-change) in abundance were represented by a Volcano plot generated in R statistical software (version 4.0.3). The statistical analysis for proteomic level was described in separated paragraph in Materials and Methods section in **(iv) Quantitative proteomics and statistical analyses and not in the paragraph of Statistical analysis**.

Line 192: What is meant by "fragile"?

Our response: Thank you for this question. "Fragile" is a term that is often used for *C. jejuni* because these bacteria are very susceptible to cultivation in the laboratory. However, we feel that this word is not necessary and we do not want to increase the ambiguity. Therefore, we have deleted this word from the manuscript and changed two sentences that previously contained the word "fragile."

First, we decided to change this sentence as follows: "This leads us to believe that *C. jejuni*, which have limited stress tolerance (starvation and oxidative stress conditions) and growth capacity, must benefit significantly from the *luxS* gene, whose role in the life cycle is well known as it regulates many phenotypes, including motility, biofilm formation, host colonization, virulence, autoagglutination, cellular adherence and invasion, oxidative stress, and chemotaxis."

Secondly, we have decided to change the first sentence in the Discussion section as follows: "*C. jejuni* which show limited hardiness and growth capabilities, remains the leading cause of foodborne bacterial gastroenteritis in humans worldwide."

Line 194-196: Vague.

Our response: Thank you for this comment, which has obviously caused some confusion, and we agree that it is also unnecessary. We have deleted the second sentence and highlighted what was the contribution of the research. Therefore, we changed the first paragraph in the Discussion section as follows: "Little is known regarding how genetic diversity and metabolic capabilities affect their metabolic phenotype and pathogenicity (27). To fill the gap regarding the role of the *luxS* gene in the *C. jejuni* life cycle, we used *C. jejuni* NCTC 11168 (wild type) and *C. jejuni* 11168 Δ *luxS* (*luxS* mutant)."

Starting Line 203: Most transcriptomic analyses in ANY organism under different conditions or a knockout etc will show differential regulation of "metabolic genes". Why do the authors think that it is noteworthy or significant that they find these genes in their analyses? Is there a way to convince the readers that these are actually important for the different phenotypes seen between wild-type and luxS mutants?

Our response: Thank you for that emphasis. We think that it is very important to highlight these genes because they explain the observations we have made at the physiological level and only for *luxS* mutant.

When the *luxS* gene is deleted, survival under starvation conditions is better because other metabolic pathways, including oxidative phosphorylation, carbon metabolism, the citrate cycle, biosynthesis of secondary metabolites, and biosynthesis of various essential amino acids, are upregulated compared with wild type. These up-regulated metabolic pathways influenced the better survival of the *luxS* mutant under starvation conditions.

We added the paragraph as follows: “Transcriptomic analysis of the *luxS* mutant, which was performed in optimal cultivation conditions (optimal temperature, atmosphere, and nutrient source), revealed up-regulation of metabolic pathways for *luxS* mutant when normalized to wild type. More in detail, when the *luxS* gene is deleted, survival under starvation conditions is better because other metabolic pathways, including oxidative phosphorylation, carbon metabolism, citrate cycle, biosynthesis of secondary metabolites, and biosynthesis of different essential amino acids. The same was determined with proteomic analysis. Those metabolic pathways are necessary for *C. jejuni* energy production and growth (28, 29). Some of the genes that are important for anabolism were up-regulated, and some of the genes that are important for catabolism of *C. jejuni* were down-regulated in *luxS* mutant. Up-regulation of genes and metabolic pathways included in anabolism can mean that *C. jejuni* synthesizes those essential amino acids itself, so this mutant does not need to take them from its environment. Down-regulation of genes included in catabolism of *C. jejuni* is then a logical consequence of up-regulation of genes included in the biosynthesis of different essential amino acids. It can be assumed that the *luxS* mutant redirects energy to the up-regulation of those other metabolic pathways, which helps the bacteria survive under various conditions. It is interesting to note that Quiñones et al. (21) have shown that the *luxS* mutant has increased chemotaxis to amino acids and decreased chemotaxis to organic acids.”

Line 242: Not sure what "a better metabolism" means?

Our response: Thank you for this highlight. We have clarified the sentence as follows: “Up-regulation of genes important for metabolism occurred in the *luxS* mutant, which indicates that the *luxS* mutant has a wider range of active metabolic pathways than the wild type during the same cultivation conditions, considering the results from the transcriptomic and proteomic analysis.”

Line 275: I am not sure what this means? "poorly utilized only 22 different carbon and nitrogen sources compared to the wild type"

Our response: Thank you for that question. We wanted to see if the *luxS* mutant would grow as well as the wild type on 288 different carbon and nitrogen substrates. In the *luxS* mutant, the s-adenosyl homocysteine recycling pathway (SAM) is disrupted and alteration of the SAM pathway is associated with significant effects on bacterial metabolism (Adler et al., 2014). This led us to predict that the *luxS* mutant will poorly metabolize some substrates, which will subsequently lead to weaker growth of the *luxS* mutant compared to the wild type. Indeed, we have shown that the *luxS* mutant poorly metabolizes 22 different carbon and nitrogen substrates, implying that the growth of the *luxS* mutant on these substrates is weaker or absent. This could be the consequence of down-regulation of some genes involved in *C. jejuni* catabolism, as shown by our results.

To clarify the goal of this analysis, we have added the following to the sentence in this paragraph as follows: “In addition, we wanted to see if the *luxS* mutant would grow as well as the wild type on 288 different carbon and nitrogen substrates. The Biolog phenotype microarray analysis revealed that *luxS* mutant poorly utilized only 22 different carbon and nitrogen sources compared to the wild type.”

Reference which we cited in our response:

Adler L, Alter T, Sharbati S, Götz G. 2014. Phenotypes of *Campylobacter jejuni luxS* mutants are depending on strain background, kind of mutation and experimental conditions. *PLOS ONE* 9:e104399.

Line 305: When is it appropriate to use kanamycin in the media. Please calrify.

Our response: Thank you for this question. MH agar was supplemented with kanamycin only in one case. We added kanamycin to MH agar when we wanted to revitalise the *C. jejuni* 11168 Δ *luxS* mutant. In the *C. jejuni* 11168 Δ *luxS* mutant, the *luxS* gene is deleted and replaced by the kanamycin cassette. Kanamycin is a selective additive that allows the growth of this mutant and inhibits the growth of cells that have lost this mutation (Elvers and Park, 2002).

We have improved following sentence as follows: “A pure culture was transferred to MH agar (Oxoid, UK) without antibiotic (wild type) or supplemented with kanamycin (30 mg/L) (Sigma Aldrich, Germany) (*luxS* mutant).”

Reference which we cited in our response:

Elvers KT, Park SF. 2002. Quorum sensing in *Campylobacter jejuni*: detection of a *luxS* encoded signalling molecule. *Microbiology*, 148, 1475–1481. <https://doi.org/10.1099/00221287-148-5-1475>

Figure 6 and 9 are confusing. I do not understand these. Why are different sections of the circle connected?

Our response: Thank you very much for this question. The following is an explanation of the answer to the question. Of course, if you think it is necessary, we can add it. At present, the article is not described in detail or as presented in other publications. However, if the examiner deems it necessary, we will add a detailed explanation to the article.

Circos supports several types of charts, such as histograms, scatter plots, and heat maps. Each Circos chart can contain multiple tracks with different sub-plots, making it ideal for visualising high-dimensional data. We used ideograms for our data, which can represent your major data classes. For genomic data, this is usually chromosomes, but it can also be species, genes, or another level of resolution depending on what relationships you want to represent. For non-genomic data, it can be individuals in a population, countries, or any other important facet of your data that you want to use for grouping. Within this ideogram, we can plot data traces. There are different plot types available, such as scatter plots, histograms, heat maps, and link tracks. In our case, we used line traces, more specifically ribbons, which show relationship between objects by a line between them. Thus, using Figure 6, we can say that the blue and pink segments represent the total number (absolute and relevant) of up-regulated and down-regulated genes.

Functional categories are represented by coloured segments:

- A - rRNA binding
- B - secondary active transmembrane transporter activity
- C - symporter activity
- D - solute:cation symporter activity
- E - solute:sodium symporter activity
- F - transporter activity
- G - ion binding
- H - cofactor binding
- I - oxidoreductase activity acting on NAD (P)H
- J - oxidoreductase activity acting on NAD (P)H, quinone or similar compound as acceptor

The number of genes upregulated is indicated by K, the number of genes downregulated is indicated by L, and no change is indicated by M. For example, the functional category G is separated into 2 ribbons. One is associated with K and the other with M, so we can see that 185 out of 410 genes belonging to functional category G are upregulated because they are associated with category K, while others have no changes in expression (they are associated with category L).

Question: How many replicates were used for RNA-seq conditions for each strain?

How many replicates were used for proteome conditions for each strain?

Our response: Thank you for that question. We wanted to see the general picture of the transcriptome and perform a deep transcriptome analysis of the *luxS* mutant and its wild type, as in our previously published article (Ramić et al., 2021). The hypotheses we had made regarding the transcriptome results were confirmed by a detailed proteomic analysis in which 3 biological replicates were performed for each strain.

Following previous publications, we have performed an in-depth analysis of the system as we envisioned it from the point of view of the overall analysis of both the transcriptome and the proteome. We would particularly like to emphasize that she stressed that it is really very important that the results come from two levels of biology - both the level of nucleic acids and proteomics.

In addition, the reviewer's comment prompted us to establish a correlation between transcriptome and proteome data, for which we are very grateful. A Pearson correlation between the transcriptomic and proteomic data is presented in Supplementary Figure S2 and confirms positive correlation between the nucleic acid level and proteomics.

We have added to the Materials and Methods section, more specifically to the Statistical Analysis subsection, as follows: To evaluate the effects of stress conditions on bacterial growth, Student's t-tests (paired) were used. "Correlation analysis using Pearson's method was performed for significantly differentially expressed results of transcriptomic and proteomic expression data." Statistical analysis was performed with IBM SPSS Statistics 23 (Statsoft Inc., Tulsa, OK, USA). A P -value < 0.05 was considered statistically significant.

We added also the Discussion section as follows: "The same was determined with proteomic analysis and Pearson's correlation test confirmed strong correlation between transcriptome and proteome data (Fig S2)."

We also modified some sentence in the Discussion section as follows:

- "Transcriptomic analysis of the *luxS* mutant, which was cultivated in optimal conditions (optimal, atmosphere, medium, and temperature)..."
- "This was also evident from the results obtained at the physiological level, where growth under stress conditions (starvation and oxidative stress) and the utilization of different carbon and nitrogen sources were tested."
- "Overall, we can say that *luxS* mutant had lower growth and survival capacity on these 22 different carbon and nitrogen sources compared to the wild type."

We added the conclusion paragraphs as follows: "To conclude, transcriptomic and proteomic analysis revealed major biological differences regarding TCA cycle and pyruvate, nitrogen, and thiamine metabolism, as well as in lipopolysaccharide biosynthesis in the *luxS* mutant compared to its wild type. Those processes were up-regulated, which could be due to the diversion of energy in the *luxS* mutant toward important processes for self-preservation. These newfound findings suggest that the *luxS* mutant is better prepared for stress conditions compared to its wild type. The *luxS* mutant survived better under starvation conditions and utilized various carbon and nitrogen sources almost as well as the wild type, which is confirmed by the results obtained at the molecular level. Moreover, the results of this manuscript significantly complement the recent and very important, finding that the concentration of AI-2 signaling molecules (the product of the *luxS* gene) increase linearly with the increment of cell concentration (17). Thus, as the first, we confirm the strong involvement of *luxS* at molecular level in the central metabolic pathway, beyond the true quorum sensing signaling system."

In addition, we added the **Supplementary Figure S2** in the Supplement document as follows:

Supplementary Figure S2. Correlation analysis using Pearson's method for transcriptomic and proteomic data, which are significantly differentially expressed in both transcriptomic and proteomic datasets. It can be concluded that proteomic and transcriptomic expression data are significantly correlated with a Pearson correlation coefficient (R) of 0.92 and a P -value of $2.094e^{-13}$ ($t = 12.477$, $df = 30$).

October 13, 2022

Dr. Anja Klančnik
University of Ljubljana
Ljubljana
Slovenia

Re: Spectrum02572-22R1 (The Role of luxS in Campylobacter jejuni Beyond Intercellular Signaling)

Dear Dr. Anja Klančnik:

Thank you for your second submission, after reading reviewer's comments and recommendation, please address all of their comments. I also recommend to carefully re-edit the writing as there was still a comment that some paragraphs are hard to read. Another point to pay attention to is the comment regarding the use of PolyA enrichment for enrichment of bacterial mRNA.

Link Not Available

Sincerely,

livnat afriat-jurnou

Journals Department
Reviewer comments:

Reviewer #2 (Comments for the Author):

Thank you for so thoroughly addressing my prior comments. I appreciate how much work was put into the revision, all of my comments were addressed.

Reviewer #3 (Comments for the Author):

I liked the comprehensive approach of cross checking the transcriptomic data with proteomic data to provide a more robust

dataset. The data has good potential however the manuscript does need significant editing as there many sentences and paragraphs that either do not read well or do not make sense, some if these sentences or paragraphs are listed in the following review. Further comparisons utilising the data already produced would enhance the manuscript, such as inclusion of tables identifying differentially expressed genes validated by proteomic analysis. Additionally further phenotypic data of a luxS complement strain would be beneficial but not essential to this data set. Given the vast list of genes identified as differentially expressed, inclusion of more stringent parameters for the identification of differentially expressed genes may provide a more robust dataset.

Line 44-46 - the statement here about stress tolerance and growth capacity is quite vague, consider rephrasing, for example to explain what is meant by limited growth capacity.

Line 60-62 - vary vague statement, needs further explanation.

Line 62-65 - this sentence does not make sense, needs to be rewritten as what you are trying to say is a valid point.

Line 67 - expand this sentence to explain survival.

Line 86 - 'presence' could be replaced with transcription or regulation etc.

Line 106 - what was the basis for determining 'optimal conditions'. Perhaps include a reference or a clearer description.

Line 135-137 - Discussion in results section. Sentence also needs re-writing.

Line 140 - can you include the exact time point used here rather than just in the methods section?

Line 141 - why were numerous changes in gene expression expected at this timepoint. Additionally what would these changes be in reference to?

Line 145 - state the fold change cut off here

Line 148 - what is meant by 'selected because they explain the observations'?

Line 156 - which two component system?

Paragraph lines 144-162 - this paragraph needs to be rewritten as it does not flow well.

Line 163 - As the list of genes identified was very vast, the data may benefit from an additional list of differentially expressed genes with a higher fold change cut off value.

Line 66-67 - is it not expressed as up or down because there was a variation in genes both up or down regulated in both strains? Could this then be an artifact of statistical parameters being too loose to accurately identify differently expressed genes. Or is it because different pathways within this GO group were up or down regulated in each strain in which case a more detailed explanation is needed here.

Line 176 or line 178 - typo, is the total number of proteins 63 or 53.

Results section - inclusion of a comparison of genes differentially expressed corresponding to proteins differentially expressed would be beneficial to this data set. Especially given the magnitude of genes identified as differentially expressed. There is a statistical analysis figure presented in the supplementary figures but there is no table or list detailing correlations between proteins and corresponding protein coding genes.

Line 201 - is it true to say 'C. jejuni shows limited hardiness' given its ability to persist in the environment to enter into the food chain? Perhaps a better description would be to describe C. jejuni as being fastidious.

Line 212 - there needs to be a reference justifying describing the culture conditions as optimal.

Line 214 - 215 - why do these metabolic pathways or how do these metabolic pathways benefit survival. Include a reference here and some examples of how these pathways could impact survival.

Line 247 - This is a significant discussion point that needs to be discussed further. Expand on the experimental differences between these studies or at least include further details of the conditions used between studies. The differing results between this study and the referenced publication needs a detailed discussion.

Line 327 - is MH media optimum? If so why or a reference for this?

Line 337-338 - were OD readings taken, given the significant discussion of VBNCs within the manuscript, the inclusion of both OD and CFU would be beneficial to this manuscript.

Line 375-376 - why was PolyA enrichment performed on prokaryotic mRNA?

Other: The data would benefit from the inclusion of a luxS complement in some of the phenotypic assays at least. E.g could it revert some of the differences observed in Figure 2.

How many biological replicates were included for RNA seq analysis?

Figure 2B - final error bar missing for luxS mutant.

Staff Comments:

Preparing Revision Guidelines

- Point-by-point responses to the issues raised by the reviewers in a file named "Response to Reviewers," NOT IN YOUR

COVER LETTER.

- Upload a compare copy of the manuscript (without figures) as a "Marked-Up Manuscript" file.
- Each figure must be uploaded as a separate file, and any multipanel figures must be assembled into one file.
- Manuscript: A .DOC version of the revised manuscript
- Figures: Editable, high-resolution, individual figure files are required at revision, TIFF or EPS files are preferred

Please return the manuscript within 60 days; if you cannot complete the modification within this time period, please contact me. If you do not wish to modify the manuscript and prefer to submit it to another journal, please notify me of your decision immediately so that the manuscript may be formally withdrawn from consideration by Microbiology Spectrum.

Reviewer #3 (Comments for the Author):

I liked the comprehensive approach of cross checking the transcriptomic data with proteomic data to provide a more robust dataset. The data has good potential however the manuscript does need significant editing as there many sentences and paragraphs that either do not read well or do not make sense, some if these sentences or paragraphs are listed in the following review. Further comparisons utilising the data already produced would enhance the manuscript, such as inclusion of tables identifying differentially expressed genes validated by proteomic analysis. Additionally further phenotypic data of a luxS complement strain would be beneficial but not essential to this data set. Given the vast list of genes identified as differentially expressed, inclusion of more stringent parameters for the identification of differentially expressed genes may provide a more robust dataset.

Line 44-46 - the statement here about stress tolerance and growth capacity is quite vague, consider rephrasing, for example to explain what is meant by limited growth capacity.

Our reply: Thank you for this suggestion. We have revised the statement in the importance section as follows:

"This led us to believe that *C. jejuni*, which is highly sensitive to stress factors (starvation and oxygen concentration) and has a low growth rate, benefits significantly from the *luxS* gene. The role of this gene in the life cycle of *C. jejuni* is well known and regulates many phenotypes, including motility, biofilm formation, host colonization, virulence, autoagglutination, cellular adherence and invasion, oxidative stress, and chemotaxis."

Line 60-62 - vary vague statement, needs further explanation.

Our reply: Thank you for this suggestion. We have corrected and elaborated the statement as follows:

"*C. jejuni* is an important bacterial species due to its high incidence in the food industry and public health system, the duration of campylobacteriosis, possible complications such as Guillain-Barré syndrome, and the wide distribution of multidrug-resistant *C. jejuni* strains. These issues not only have an impact on food safety and healthcare but also on socioeconomics. Campylobacters are responsible for over 166 billion gastrointestinal diseases and approximately 38 000 deaths per year worldwide (Gözl et al., 2018)."

Reference:

Gözl G, Kittler S, Malakauskas M, Alter T. 2018. Survival of *Campylobacter* in the food chain and the environment. *Curr Clin Microbiol Rep* 5:126–134.

Line 62-65 - this sentence does not make sense, needs to be rewritten as what you are trying to say is a valid point.

Our reply: Thank you for this comment. We have revised our statement as follows:

"Moreover, although *C. jejuni* is difficult to cultivate in the laboratory, it is a widespread and persistent bacteria in the food industry; this is known as the "*Campylobacter paradox*"."

Line 67 - expand this sentence to explain survival.

Our reply: Thank you for this suggestion. We have revised and expanded the statement as follows:

“In addition, temperature fluctuations, oxidative stress, starvation, and other stress factors present in the food-processing environment trigger the transition to a viable but non-culturable state (VBNC), which correlates with prolonged persistence of *Campylobacter* in the food chain. In this state, *C. jejuni* can still infect humans.”

Line 86 - 'presence' could be replaced with transcription or regulation etc.

Our reply: Thank you for this suggestion. We changed “presence” to “transcription”.

Line 106 - what was the basis for determining 'optimal conditions'. Perhaps include a reference or a clearer description.

Our reply: Thank you for this comment. We have now cited a reference (see below), in which optimal conditions for *C. jejuni* growth are described.

Reference:

Davis L, DiRita V. 2008. Growth and laboratory maintenance of *Campylobacter jejuni*. *Curr Protoc Microbiol* 10:8A.1.1-8A.1.7.

Line 135-137 - Discussion in results section. Sentence also needs re-writing.

Our reply: Thank you for this comment. We have re-written the sentence as follows:

“Transcriptomic profile of the *luxS* mutant and wild type. Comparing the transcriptomes between wild type and mutant *luxS* can improve our understanding of the role of the *luxS* gene in the life cycle of *C. jejuni*. Strains were cultivated until the middle of the exponential phase, during which numerous changes in gene expression are expected.”

Line 140 - can you include the exact time point used here rather than just in the methods section?

Our reply: Thank you for this comment. We have added the requested information to the sentence as follows:

“Detailed analysis showed that 765 genes were differentially expressed in the *luxS* mutant when normalized to the wild type (FDR, $P \leq 0.05$) after 16 h of incubation.”

Line 141 - why were numerous changes in gene expression expected at this timepoint. Additionally what would these changes be in reference to?

Our reply: Thank you for these questions. The selected time point was chosen because it is known that bacteria divide and actively grow in the exponential phase. At this time point, most genes are expressed, especially genes involved in bacterial metabolism. Bacterial metabolism must be strongly activated because cells need new and diverse building materials to form complete cells. The middle exponential phase (16 h of incubation) was used to detect differentially expressed genes in the *luxS* mutant compared to the wild type, as previously described by Ramić et al. (2021).

Reference:

Ramić D, Bucar F, Kunej U, Dogša I, Klančnik A, Smole Možina S. 2021. Antibiofilm potential of *Lavandula* preparations against *Campylobacter jejuni*. *Appl Environ Microbiol* 19:e0109921.

Line 145 - state the fold change cut off here

Our reply: Thank you for this comment. We have added the requested information in the sentence as follows:

“However, the list (Supplementary Table 1) of these transcripts was selected because they significantly differed between the *luxS* mutant and wild type (\log_2 fold-change ≥ 1 , $P \leq 0.05$).”

Line 148 - what is meant by 'selected because they explain the observations'?

Our reply: Thank you for this question. We have revised and elaborated this statement as follows:

“Some of these genes are listed below because they can explain our observations at the physiological level, such as growth under starvation-like conditions.”

Line 156 - which two component system?

Our reply: Thank you for this question. We have added which two-component systems were meant:

“Many other pathways, including two-component systems, such as flagellar assembly, ABC transporters, and protein transport, were also up-regulated, whereas many ribosomal genes were down-regulated.”

Paragraph lines 144-162 - this paragraph needs to be rewritten as it does not flow well.

“Detailed analysis showed that 765 genes were differentially expressed in the *luxS* mutant when normalized to the wild type (FDR, $P \leq 0.05$) after 16 h of incubation. Among them, 354 genes were up-regulated, and 402 genes were down-regulated (Supplementary Table 1; Fig. 5). However, the list (Supplementary Table 1) of these transcripts was selected because they significantly differed between the *luxS* mutant and wild type (\log_2 fold-change ≥ 1 , $P \leq 0.05$). Some of these genes are listed below because they can explain our observations at the physiological level, such as growth under starvation-like conditions. Among the up-regulated genes, genes important for *C. jejuni* anabolism were detected, including *lys-C*, *aspS*, *pyrB*, *glnP*, *glnS*, *proA*, *gltB*, *argC*, *hemA*, and *ilvI*. Among the down-regulated genes, genes important for *C. jejuni* catabolism were detected, including *htrA*, *pyk*, *Cj0021c*, *Cj1418c*, *Cj1417c*, *purQ*, *aspB*, *Cj0073c*, and *fcl* (Table 1). The network of differentially expressed genes in the *luxS* mutant showed that most of the genes are included in biological processes, such as the TCA cycle; the metabolism of pyruvate, nitrogen, and thiamine; and lipopolysaccharide biosynthesis, and that these biological processes were up-regulated. Many other pathways, including two-component systems, such as flagellar assembly, ABC transporters, and protein transport, were also up-regulated, whereas many ribosomal genes were down-regulated. (Fig. 6). Of 30 differentially expressed genes involved in flagellar motility and the colonization of abiotic and biotic surfaces, 22 were up-regulated: *flhF*, *flgI*, *flgB*, *flaG*, *flgM*, *flgD*, *flgG2*, *flgE2*, *fliD*, *flgK*, *fliS*, *flaA*, *pseB*, *pseC*, *ptmA*, *maf4*, *legF*, *Cj1319*, *cj1330*, *cj1026c*, *Cj0391c*, and *Cj0977*. Furthermore, numerous genes important for the stress response were either up-regulated (*cstA*, *tpx*, *trxA*, *trxB*, *kata*, *fdxA*, and *ahpC*) or down-regulated (*grpE*, *dnaK*, and *htpG*).”

Line 163 - As the list of genes identified was very vast, the data may benefit from an additional list of differentially expressed genes with a higher fold change cut off value.

Our reply: Thank you for this suggestion. We used standard fold-change cut-off values that we wanted to keep constant to compare the proteomic and transcriptomic data. We also referred to a previously published article by Ramić et al. (2021) in which the same cut-off values were used. We have added a summary table below with all genes with an absolute \log_2 fold change ≥ 2.5 and a P -value ≤ 0.05 . For better clarity of the Supplementary table 1, we have highlighted all genes in grey with an absolute \log_2 fold change ≥ 2.5 and added additional information in the caption as stated below:

“SUPPLEMENTARY TABLE 1 Differentially expressed genes in *C. jejuni* 11168 Δ *luxS* normalized to the *C. jejuni* NCTC 11168 (FDR, $P \leq 0.05$). Genes with an absolute \log_2 fold change ≥ 2.5 are highlighted in grey.”

Summary table 1: Differentially expressed genes in *C. jejuni* 11168 Δ *luxS* normalized to the *C. jejuni* NCTC 11168 (FDR, $P \leq 0.05$) with \log_2 fold change ≥ 2.5 .

Gene name	Log ₂ fold-change	False discovery rate P -value
Down-regulated		
Cj0633	-2.5	0E + 00
tatA	-2.5	0E + 00
Cj1623	-2.5	0E + 00
Cj1680c	-2.5	0E + 00
rplF	-2.5	0E + 00
exbB3	-2.6	0E + 00
clpB	-2.6	0E + 00
Cj0892c	-2.6	0E + 00
Cj1004	-2.6	0E + 00
Cj1089c	-2.6	0E + 00
metF	-2.6	0E + 00
rpsM	-2.6	0E + 00
Cj0374	-2.7	0E + 00
Cj0539	-2.7	0E + 00
ssb	-2.7	0E + 00
Cj1191c	-2.7	0E + 00
rpoA	-2.7	0E + 00
cbf2	-2.8	0E + 00
Cj0776c	-2.8	0E + 00
Cj0854c	-2.8	0E + 00
Cj0993c	-2.8	0E + 00
rplE	-2.8	0E + 00
rpsC	-2.8	0E + 00
atpH	-2.9	0E + 00
Cj0397c	-2.9	0E + 00

grpE	-2.9	0E + 00
Cj1380	-2.9	0E + 00
Cj1041c	-3.0	0E + 00
dnaN	-3.1	0E + 00
Cj0036	-3.1	0E + 00
Cj0421c	-3.1	0E + 00
mobA	-3.1	0E + 00
Cj1459	-3.1	0E + 00
Cj1483c	-3.1	0E + 00
Cj1621	-3.1	0E + 00
rplR	-3.1	0E + 00
rpsQ	-3.1	0E + 00
rpsF	-3.2	0E + 00
Cj1412c	-3.2	0E + 00
Cj1381	-3.3	0E + 00
Cj0152c	-3.4	0E + 00
tig	-3.4	0E + 00
Cj0573	-3.4	0E + 00
Cj0459c	-3.5	0E + 00
rplI	-3.5	0E + 00
rimM	-3.5	0E + 00
Cj1062	-3.5	0E + 00
Cj0406c	-3.6	0E + 00
Cj1036c	-3.9	0E + 00
Cj1533c	-4.2	0E + 00
Cj0717	-4.3	0E + 00
Cj0040	-4.5	0E + 00
Cj0114	-4.8	0E + 00
Cj0143c	-4.8	0E + 00
tatB	-4.8	0E + 00

Cj0323	-5.0	0E + 00
Cj1000	-6.3	0E + 00
Cj1078	-3.8	1E - 02
bioC	-2.6	1E - 03
Cj0544	-3.5	1E - 03
Cj0120	-2.5	1E - 04
Cj0730	-3.4	1E - 04
Cj0376	-3.8	1E - 09
Cj0341c	-3.2	1E - 10
Cj0058	-3.0	1E - 11
Cj0520	-4.5	1E - 11
folB	-2.6	1E - 13
Cj1417c	-2.7	1E - 13
Cj1460	-2.8	1E - 14
Cj0839c	-2.6	2E - 02
aroK	-3.4	2E - 03
Cj0177	-4.5	2E - 03
Cj1232	-3.4	2E - 04
Cj1162c	-5.4	2E - 04
moaD	-3.1	2E - 06
Cj0488	-3.4	2E - 10
rpmC	-4.0	2E - 10
leuC	-4.5	2E - 11
Cj1496c	-4.0	2E - 13
hemN	-2.5	2E - 15
Cj0948c	-2.8	3E - 03
Cj1383c	-4.4	3E - 03
Cj0620	-3.3	3E - 05
Cj0090	-3.3	3E - 05
Cj0038c	-3.2	3E - 08

nssR	-5.1	3E - 14
Cj0395c	-9.9	4E - 02
Cj1028c	-2.5	4E - 07
tonB1	-4.1	4E - 08
Cj1042c	-3.7	4E - 09
Cj0124c	-4.7	4E - 10
Cj1340c	-4.1	4E - 11
Cj1063	-5.6	4E - 11
Cj0261c	-2.5	4E - 16
rplW	-3.1	4E - 16
Cj0829c	-5.0	5E - 04
Cj1589	-6.5	5E - 06
Cj0823	-3.2	5E - 07
Cj0732	-2.5	6E - 03
Cj0416	-2.5	6E - 04
feoA	-3.2	6E - 08
Cj0881c	-4.4	6E - 13
purS	-6.5	7E - 06
kdtA	-2.8	7E - 08
selB	-2.5	7E - 13
nusB	-3.3	7E - 14
Cj0162c	-3.8	7E - 15
Cj1301	-2.7	8E - 11
Cj1021c	-2.5	8E - 16
Cj0510c	-3.0	8E - 16
Cj0251c	-3.9	9E - 03
cgb	-2.8	9E - 05
Up-regulated		
ahpC	2.5	0E + 00
Cj0667	2.5	0E + 00

hup	2.5	0E + 00
thiG	2.5	0E + 00
Cj1112c	2.5	0E + 00
pseA	2.5	0E + 00
legF	2.5	0E + 00
Cj0935c	2.6	0E + 00
pseB	2.6	0E + 00
Cj0069	2.7	0E + 00
cobB	2.7	0E + 00
pglK	2.7	0E + 00
Cj1164c	2.7	0E + 00
porA	2.7	0E + 00
Cj1548c	2.7	0E + 00
atpE	2.8	0E + 00
gmhA1	2.8	0E + 00
atpB	2.8	0E + 00
rpsL	2.9	0E + 00
zupT	3	0E + 00
fdxB	3	0E + 00
Cj0391c	3	0E + 00
glnP	3	0E + 00
Cj1656c	3	0E + 00
sdhB	3.1	0E + 00
Cj1241	3.1	0E + 00
hydC	3.1	0E + 00
Cj1555c	3.1	0E + 00
accB	3.2	0E + 00
pstA	3.3	0E + 00
Cj0593c	3.5	0E + 00
hydA	3.9	0E + 00

petA	4.3	0E + 00
Cj1169c	4.5	0E + 00
Cj0035c	3.1	4E - 05

Reference:

Ramić D, Bucar F, Kunej U, Dogša I, Klančnik A, Smole Možina S. 2021. Antibiofilm potential of *Lavandula* preparations against *Campylobacter jejuni*. *Appl Environ Microbiol* 19:e0109921.

Line 66-67 - is it not expressed as up or down because there was a variation in genes both up or down regulated in both strains? Could this then be an artifact of statistical parameters being too loose to accurately identify differently expressed genes. Or is it because different pathways within this GO group were up or down regulated in each strain in which case a more detailed explanation is needed here.

Our reply: Thank you for this question. The statistical parameters were defined as in our previous work (Ramić et al., 2021). We wanted to gain a wider picture of the genes involved in different pathways in the *C. jejuni* 11168Δ*luxS* mutant to determine which other pathways are important for the metabolism of the *luxS* mutant and whether these pathways are more or less pronounced in the *luxS* mutant. Genes inside the same GO group were down- and up-regulated at equal ratios (as shown in the Figure below):

This scheme depicts functional categorization of molecular functions in *C. jejuni* 11168 Δ luxS (*luxS* mutant). Blue and pink outer segments represent the total number of up- and down-regulated genes, respectively. Functional categories are represented by segments (A: rRNA binding; B: secondary active transmembrane transporter activity; C: symporter activity; D: solute:cation symporter activity; E: solute:sodium symporter activity; F: transporter activity; G: ion binding; H: cofactor binding; I: oxidoreductase activity, acting on NAD(P)H; J: oxidoreductase activity, acting on NAD(P)H, quinone, or a similar compound as the acceptor. K, L, and M indicate the number of genes that were up-regulated, down-regulated, or unaltered, respectively.

This Figure is part of Figure 7 in the manuscript.

Reference:

Ramić D, Bucar F, Kunej U, Dogša I, Klančnik A, Smole Možina S. 2021. Antibiofilm potential of *Lavandula* preparations against *Campylobacter jejuni*. *Appl Environ Microbiol* 19:e0109921.

Line 176 or line 178 - typo, is the total number of proteins 63 or 53.

Our reply: Thank you for this comment. We have corrected the number of proteins to 53 in the sentence.

Line 201 - is it true to say 'C. jejuni shows limited hardiness' given its ability to persist in the environment to enter into the food chain? Perhaps a better description would be to describe C. jejuni as being fastidious.

Our reply: Thank you for this comment. We have substituted the word “limited hardiness” with “fastidious” in the sentence, which was further expanded as follows:

“Even though *C. jejuni* is a fastidious bacterium with limited growth capabilities outside of the host, it is widespread in the food-processing environment and remains the leading cause of foodborne bacterial gastroenteritis in humans worldwide.”

Line 212 - there needs to be a reference justifying describing the culture conditions as optimal.

Our reply: Thank you for this comment. We have added the appropriate reference at the end of the sentence:

“Transcriptomic analysis of the *luxS* mutant, which was performed under optimal cultivation conditions (with parameters regarding atmosphere, medium, and temperature according to Davis and DiRita (2008)), revealed up-regulation of metabolic pathways for the *luxS* mutant when normalized to wild type.”

Reference:

Davis L, DiRita V. 2008. Growth and laboratory maintenance of *Campylobacter jejuni*. *Curr Protoc Microbiol* 10:8A.1.1-8A.1.7.

Line 214 - 215 - why do these metabolic pathways or how do these metabolic pathways benefit survival. Include a reference here and some examples of how these pathways could impact survival.

Our reply: Thank you for this comment. The *luxS* gene plays many roles that are not yet fully understood. *LuxS* is not only involved in intercellular signalling but also in many other biological processes, such as essential amino acid synthesis, nitrogen and pyruvate metabolism, and flagellar assembly. Deletion of the *luxS* gene changes the metabolism of *C. jejuni*, promoting synthesis of new essential molecules, such as amino acids that contribute to survival.

We have revised the sentence and added new references:

Specifically, when the *luxS* gene is deleted, survival under starvation conditions is better because other metabolic pathways are up-regulated, including oxidative phosphorylation, carbon metabolism, citrate cycle, and the biosynthesis of secondary metabolites and different essential amino acids (Shahrezaei and Marguerat, 2015; Klumpp and Hwa, 2014). Veselovsky et al. (2022) stated that during the exponential phase, synthesis of different essential amino acids encourages cell growth.

References:

Shahrezaei V, Marguerat S. 2015. Connecting growth with gene expression: of noise and numbers. *Curr Opin Microbiol* 25:127–135.

Klumpp S, Hwa T. 2014. Bacterial growth: global effects on gene expression, growth feedback and proteome partition. *Curr Opin Biotechnol* 28:96–102.

Veselovsky VA, Dyachkova MS, Bespiatykh DA, Yunes RA, Shitikov EA, Polyayeva PS, Danilenko VN, Olekhovich EI, Klimina KM. 2022. The gene expression profile differs in growth phases of the *Bifidobacterium longum* culture. *Microorganisms* 10: 1683.

Line 247 - This is a significant discussion point that needs to be discussed further. Expand on the experimental differences between these studies or at least include further details of the conditions used between studies. The differing results between this study and the referenced publication needs a detailed discussion.

Our reply: Thank you for this comment. In our experimental approach, we used 5% of *C. jejuni* $\Delta luxS$ overnight culture to prepare the experimental culture, whereas Holmes et al. (2009) used 3% inoculum. Furthermore, we prepared our strains in MH broth, whereas Holmes et al. (2009) used minimum essential medium. An important difference is also the timing of cell harvest (after 8 h in Holmes et al. (2009) versus 16 h in our study), which significantly influences gene expression. Cells in the early exponential phase are more susceptible to stress compared to cells in the middle or late exponential phase (Reichert-Schwillinsky et al., 2009).

To make it clearer, we have extended the sentence as follows:

“Holmes et al. (24) observed that many genes important for major stress responses were down-regulated in the *C. jejuni luxS* mutant, which is contrary to our results. However, these studies were performed under different experimental conditions and time points of cell harvest for transcriptomic analysis.”

References:

Holmes K, Tavender TJ, Winzer K, Wells JM, Hardie KR. 2009. AI-2 does not function as a quorum sensing molecule in *Campylobacter jejuni* during exponential growth *in vitro*. *BMC Microbiol* 9:1–11.
Reichert-Schwillinsky F, Pin C, Dzieciol M, Wagner M, Hein I. 2009. Stress- and growth rate-related differences between plate count and real-time PCR data during growth of *Listeria monocytogenes*. *Appl Environ Microbiol* 75:7.

Line 327 - is MH media optimum? If so why or a reference for this?

Our reply: Thank you for this comment. We have added the appropriate reference at the end of the sentence:

“Optimal conditions included a microaerobic 315 atmosphere (5% O₂, 10% CO₂, 85% N₂), optimal medium for cultivation (MH medium), and optimal 316 temperature (42 °C) (Davis and DiRita, 2008).

Reference:

Davis L, DiRita V. 2008. Growth and laboratory maintenance of *Campylobacter jejuni*. *Curr Protoc Microbiol* 10:8A.1.1-8A.1.7.

Line 337-338 - were OD readings taken, given the significant discussion of VBNCs within the manuscript, the inclusion of both OD and CFU would be beneficial to this manuscript.

Our reply: Thank you for this comment. The correlation between OD₆₀₀ and CFU/mL was achieved based on our experiments. The OD₆₀₀ value of 0.1 corresponds to $(1.1 \pm 0.1) \times 10^7$ CFU/mL. An additional sentence and additional Supplementary Table S2 were added:

Lines 326–327: “Cultures of the *luxS* mutant and wild type were adjusted at the optical density of 600 nm (OD₆₀₀) to the value of 0.1 a.u. which corresponds to $(1.1 \pm 0.1) \times 10^7$ CFU/mL (Supplementary Table S2).”

Supplementary Information Lines 124–127: **Supplementary Table S2. Measured values of OD₆₀₀ and CFU/mL for *C. jejuni* 81-176 overnight cultures. Cultures were incubated in MH broth in a microaerobic atmosphere (85% N₂, 5% O₂, 10% CO₂) at 42 °C for 24 h.**

Sample	OD600	CFU/mL
1	0.101	1.10E+07
2	0.103	1.00E+07
3	0.099	1.20E+07
4	0.101	1.00E+07
5	0.102	1.30E+07

Line 375-376 - why was PolyA enrichment performed on prokaryotic mRNA?

Our reply: Thank you for this important question. Most RNA molecules in bacteria are rRNA, and thus it is common practise to remove rRNA from the total RNA sample, so that the reads in an RNA-seq experiment are predominantly from mRNA. There are many expensive commercial kits on the market that are commonly used for this procedure. In our study, we chose a method that had already been tested in the study by Ribič et al. (2020) and gave reliable results.

Reference:

Ribič U, Jakše J, Toplak N, Koren S, Kovač M, Klančnik A, Jeršek B. 2020. Transporters and efflux pumps are the main mechanisms involved in *Staphylococcus epidermidis* adaptation and tolerance to didecyltrimethylammonium chloride. *Microorganisms* 8:344.

Other: The data would benefit from the inclusion of a luxS complement in some of the phenotypic assays at least. E.g could it revert some of the differences observed in Figure 2. How many biological replicates were included for RNA seq analysis? Figure 2B - final error bar missing for luxS mutant.

Our reply: Thank you for this comment. We wanted to see the general picture of the transcriptome and thus performed a deep transcriptome analysis of the *luxS* mutant and its wild type, as in our previously published study (Ramić et al., 2021). Our hypotheses regarding the transcriptome results were confirmed by a detailed proteomic analysis in which three biological replicates were performed for each strain.

Following previous publications, we performed an in-depth analysis of the system as we envisioned it from the point of view of the overall analysis of both the transcriptome and the proteome. We would particularly like to emphasize that it is very important that the results come from two levels of biology—both the levels of nucleic acids and proteins.

In addition, the reviewer's comment prompted us to investigate any correlations between the transcriptomic and proteomic data, for which we are very grateful. A Pearson correlation between the transcriptomic and proteomic data is presented in Supplementary Figure S2 and confirms a positive correlation between the nucleic acids and proteins.

We have added the following to the Materials and Methods section (more specifically to the Statistical Analysis subsection):

“To evaluate the effects of stress conditions on bacterial growth, Student's t-tests (paired) were used. Correlation analysis using Pearson's method was performed for significantly differentially

expressed results of transcriptomic and proteomic expression data. Statistical analysis was performed with IBM SPSS Statistics 23 (Statsoft Inc., Tulsa, OK, USA). A P -value of ≤ 0.05 was considered statistically significant.”

We added a concluding paragraph as follows:

“To conclude, transcriptomic and proteomic analysis revealed major biological differences regarding the TCA cycle; the metabolism of pyruvate, nitrogen, and thiamine; and lipopolysaccharide biosynthesis between the *luxS* mutant and wild type. These processes were up-regulated, possibly for the self-preservation of the *luxS* mutant. These novel findings suggest that the *luxS* mutant is better prepared for stress conditions compared to its wild type. The *luxS* mutant survived better under starvation conditions and utilized various carbon and nitrogen sources almost as well as the wild type. Moreover, the results of this manuscript significantly complement the recent and very important finding that the concentration of AI-2 signalling molecules (the product of the *luxS* gene) increase linearly with increasing cell concentration (17). Thus, we are the first to confirm the strong involvement of *luxS* in the central metabolic pathway, beyond the true quorum sensing signalling system.”

In addition, we added **Supplementary Figure S2** in the Supplementary document as follows:

Supplementary Figure S2. Correlation analysis using Pearson's method was performed for the data on transcript and protein expression, which are significantly differentially expressed in both transcriptomic and proteomic datasets and show the same direction of expression levels ($n=28$). The P -value of the test is $9.518e-07$, which is below the significance level of $\alpha = 0.05$. It can be concluded that proteomic and transcriptomic expression are significantly correlated with a Pearson's correlation coefficient (R) of 0.78 and a P -value of $9.518e-07$ ($t = 6.3711$, $df = 26$).

Results section - inclusion of a comparison of genes differentially expressed corresponding to proteins differentially expressed would be beneficial to this data set. Especially given the magnitude of genes identified as differentially expressed. There is a statistical analysis figure presented in the supplementary figures but there is no table or list detailing correlations between proteins and corresponding protein coding genes.

Our reply: Our response to this comment can be found in our response to the previous comment. We have added a supplementary figure and supplementary table in the supplementary document as follows:

Supplementary Figure S2. Correlation analysis using Pearson's method was performed for the data on transcript and protein expression, which are significantly differentially expressed in both transcriptomic and proteomic datasets and show the same direction of expression levels (n=28). The *P*-value of the test is 9.518e-07, which is below the significance level of alpha = 0.05. It can be concluded that proteomic and transcriptomic expression are significantly correlated with a Pearson's correlation coefficient (*R*) of 0.78 and a *P*-value of 9.518e-07 (*t* = 6.3711, *df* = 26).

Supplementary Table S3. Common differentially expressed genes and proteins in *C. jejuni* 11168Δ*luxS* compared to *C. jejuni* NCTC 11168.

Gene name	Difference in gene expression	P -value	Protein name	Difference in protein expression	P -value
rplL	2.26247431	0.00E+00	RpLL	6.147657235	9.15E-05
porA	2.740632462	0.00E+00	PorA	2.501590302	3.84E-02

folE	0.643668434	1.04E-04	FolE	2.438784117	2.87E-03
dapB	1.832144943	0.00E+00	DapB	2.252253175	2.59E-04
ctb	0.970918119	6.29E-06	Ctb	2.189983696	1.06E-02
eno	0.760573722	3.47E-10	Eno	2.028894588	3.30E-02
rpoB	0.811037856	1.86E-11	RpoB	2.0190202	7.72E-03
thiG	2.501233072	0.00E+00	ThiG	2.017311265	1.95E-03
purA	1.624247861	0.00E+00	PurA	1.646993538	2.62E-02
ilvC	1.252092764	0.00E+00	IlvC	1.637678504	5.52E-03
guaA	0.329511776	2.56E-02	GuaA	1.601129413	2.06E-02
dapE	2.357275576	0.00E+00	DapE	1.537709634	2.10E-04
leuS	0.462751733	4.34E-04	LeuS	1.496526043	5.29E-03
pseB	2.57038766	0.00E+00	PseB	1.388117433	1.93E-02
aspS	1.633461656	0.00E+00	AspS	1.310820897	2.00E-02
rplT	0.952788717	6.72E-14	RplT	1.297215263	1.09E-02
fabH	1.285473419	0.00E+00	FabH	1.181837161	7.81E-03
dapD	0.296090917	3.04E-02	DapD	1.00121816	4.57E-02
thrS	0.547612082	2.33E-05	ThrS	0.973991315	2.33E-03
hisF1	1.733245891	0.00E+00	HisF1	0.709415674	2.52E-02
cysM	0.353371487	7.33E-03	CysM	0.504441047	3.73E-02
dnaK	-1.488414134	0.00E+00	DnaK	-0.84844477	3.11E-02
recN	-1.282976373	1.79E-11	RecN	-1.096624056	4.57E-02
tupA	-0.296968014	3.37E-02	TupA	-1.195404065	3.58E-02
lpxB	-2.054545385	0.00E+00	LpxB	-1.372042815	7.57E-04
hypA	-1.811222975	6.79E-07	HypA	-1.475221713	2.71E-02
pepA	-0.416597016	4.28E-03	PepA	-1.931097984	1.32E-02
Cj1418c	-1.448429317	0.00E+00	Cj1418c	-2.215515892	1.48E-02

December 23, 2022

Dr. Anja Klančnik
Univerza v Ljubljani
Ljubljana
Slovenia

Re: Spectrum02572-22R2 (The Role of luxS in Campylobacter jejuni Beyond Intercellular Signaling)

Dear Dr. Anja Klančnik:

Your manuscript has been accepted, and I am forwarding it to the ASM Journals Department for publication. You will be notified when your proofs are ready to be viewed.

Sincerely,

livnat afriat-jurnou
Editor, Microbiology Spectrum
